# Unified momentum model for rotor aerodynamics across operating regimes

Jaime Liew [1], Kirby S. Heck [1] & Michael F. Howland [1] ✉

Despite substantial growth in wind energy technology in recent decades, aerodynamic modeling of wind turbines relies on momentum models derived in the late 19th and early 20th centuries, which are well-known to break down under flow regimes in which wind turbines often operate. This gap in theoretical modeling for rotors that are misaligned with the inflow and also for high-thrust rotors has resulted in the development of numerous empirical corrections which are widely applied in textbooks, research articles, and open-source and industry design codes. This work reports a Unified Momentum Model to efficiently predict power production, thrust force, and wake dynamics of rotors under arbitrary inflow angles and thrust coefficients without empirical corrections. The Unified Momentum Model is additionally coupled with a blade element model to enable blade element momentum modeling predictions of wind turbines in high thrust coefficient and yaw misaligned states without using corrections for these states. This Unified Momentum Model can form a new basis for wind turbine modeling, design, and control tools from first principles and may enable further development of innovations necessary for increased wind production and reliability to respond to 21st century climate change challenges.

To meet mid-century global net-zero carbon emissions targets, wind energy capacity is estimated to require between a 9- and 11-fold increase[1,2]. Within the U.S. alone, as much as a 28-fold increase in wind power capacity is required to achieve net-zero by 2050[3]. This unprecedented scale-up must be guided by modeling tools that are sufficiently accurate for the design and control of wind turbines and wind farms[4,5]. Despite substantial growth in wind energy technology in recent decades[6], aerodynamic modeling of wind turbine rotors relies heavily on models originally derived in the late 19th and early 20th centuries[7–9], which are well-known to break down in flow regimes that modern wind turbines often operate within[10–13]. To overcome discrepancies associated with this fundamental breakdown, the predictions of wind turbine forces and power output that drive contemporary design and control protocols are based on empirical formulas[14,15]. Besides numerically intensive computational fluid dynamics (CFD) simulations that have limited utility in high-throughput optimization applications, there is no existing first-principles theory that can accurately predict rotor aerodynamics across the range of thrust forces and misalignment angles commonly encountered by wind turbines. This work develops a Unified Momentum Model for rotor aerodynamics that is valid across operating regimes, from low to high thrust coefficient magnitudes, including positive and negative thrust, and at arbitrary misalignment angles between inflow and rotor. The Unified Momentum Model overcomes limitations of classical one-dimensional momentum theory by accounting for both misalignment between the rotor and inflow and the pressure deficit in the rotor wake, as predicted by a solution to the differential Euler equations. The resulting aerodynamic model predicts rotor thrust, power, and wake velocities at arbitrary misalignments and thrust coefficients without empirical corrections. This Unified Momentum Model represents a departure from traditional one-dimensional momentum theory aided with empirical corrections, offering a first-principles foundation that can serve as a new basis for modeling, designing, and controlling wind turbines.

One-dimensional momentum modeling, originally derived in the late 19th century by Rankine (1865)[7], W. Froude (1878)[8], and R.E.

[1]Civil and Environmental Engineering, Massachusetts Institute of Technology, Cambridge, MA, USA. ✉e-mail: mhowland@mit.edu

Froude (1889)[9], is the predominant model used in engineering analysis and design of rotors including wind turbines, propellers, helicopters, drones, and hydrokinetic turbines[11,16–22]. The theory, which is the starting point for any textbook on rotor aerodynamics across engineering applications[23], represents the rotor as a porous actuator disk that imparts a thrust force on the surrounding flow. The induced velocities generated by the rotor thrust are related to the upstream and downstream velocities via the conservation of mass and momentum in one dimension normal to the disk. One-dimensional momentum modeling provides reasonably accurate predictions of rotor performance at low to moderate thrust coefficients, depicted in Fig. 1 as the *windmill state*. However, the theory is well known to break down at higher thrust coefficients as well as in situations of misalignment between the inflow and the rotor, which is commonly encountered in practice[11,24]. In these regimes, the one-dimensional momentum theory exhibits high error for critical quantities including rotor thrust, power, wake velocities, and outlet pressure, demonstrating quantitative and qualitative deviation from measurements. Given the one-dimensionality of the classical model, effects caused by misalignment between the rotor and the inflow are not captured. Experiments and CFD simulations show that thrust continues to increase as induction increases[10,24,25], whereas classical momentum modeling is unable to capture this behavior, predicting the opposite trend (see Fig. 1). The discrepancies in the classical momentum theory arise from two main limiting assumptions: (1) one-dimensional flow perpendicular to the rotor and (2) recovery of wake pressure to the freestream value.

The assumption of one-dimensional flow neglects all lateral velocities induced by the rotor misalignment[26]. Wind turbines continuously operate in some degree of yaw misalignment with respect to the incident wind direction due to a slowly reacting yaw controller and error or bias in wind direction measurements[27]. These misalignment errors are predicted to be larger for floating offshore turbines[28], which are anticipated to account for a large fraction of future U.S. offshore wind energy generation[29]. One-dimensional momentum modeling is typically adjusted using empirical skewed wake corrections to represent the influence of rotor misalignment[11,14,17]. Textbooks state that the power production of a rotor yaw misaligned at angle $\gamma$ scales with $\cos^3(\gamma)$[11], while empirical observations and CFD output do not exhibit this relationship, instead showing sub-cubic behavior which varies with rotor operating conditions[30–32]. In tandem, given the proliferation of wind energy and the densification of wind farms that leads to unfavorable aerodynamic wake interactions between neighboring turbines[33,34], research has focused on methods to collectively operate turbines within a farm by controlling the wind flow to maximize farm power production[5]. The primary wind farm flow control methodology, termed wake steering, entails intentionally yaw misaligning wind turbines with respect to the freestream wind[35,36], a misalignment that results in an explicit breakdown of the one-dimensional momentum modeling used to predict the power, loads, and wake velocities associated with the yawed turbine.

The assumption that the wake pressure recovers to the freestream value is violated at higher thrust coefficients. At higher thrust coefficients, the static pressure in the wake downstream of the rotor fails to return to the freestream pressure. This persistent pressure drop behind the rotor, known as base suction[37], corresponds to an additional thrust force contribution that is not captured by current theoretical models. Specifically, classical momentum modeling is widely understood and accepted to break down at a value of the induction factor ($a = 1 - u_d/u_\infty$, where $u_d$ is the velocity at the rotor disk and $u_\infty$ is the freestream wind speed) that is only 10% higher than the optimal value of $a = 1/3$[23,38] predicted by Betz (1920)[11,25,38], based on classical one-dimensional momentum modeling. There is no comprehensive theoretical or analytical model rooted in first-principles that can adequately capture the effects caused by rotor misalignment and wake pressure. This gap in theoretical modeling has prompted the development of numerous empirical corrections, which have found widespread application in textbooks[11,39], research articles[15,21,40], and open-source, as well as industry, design codes[14,41,42]. Examples include the empirical Glauert correction to model high thrust operation[10–15,21,40], skewed wake corrections[11,14,17], and an empirically tuned power-yaw formula to model rotor misalignment[31,35,43].

This work extends first-principles understanding and develops a new analytical aerodynamic model for rotors operating in arbitrary thrust and misalignment conditions. To address long-standing discrepancies between classical momentum modeling and empirical observations, the actuator disk model is re-examined from first principles. Using conservation of mass, momentum, and energy, the limiting assumptions of the classical theory are eliminated by modeling the pressure deficit in the rotor wake, using a solution to the differential Euler equations, and by accounting for arbitrary rotor misalignment with a lifting line model. This results in a set of five coupled equations that simultaneously govern rotor induction, thrust, wake

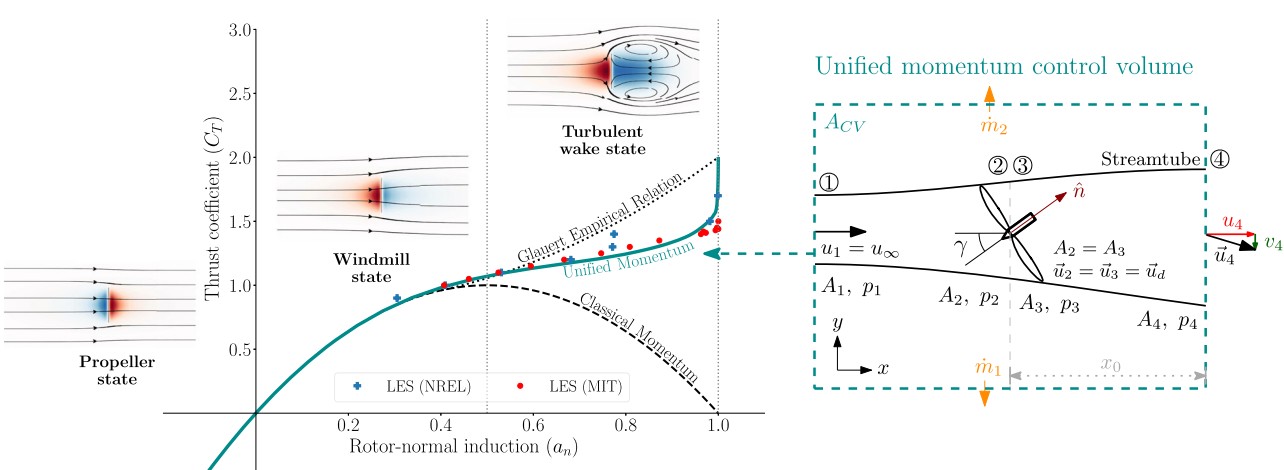

**Fig. 1 | Schematic of the regimes of rotor operation and the Unified Momentum Model.** (Left) Schematic illustrating the rotor thrust coefficient variations with rotor-normal induction across various operational scenarios (propeller state, windmill state, and turbulent wake state[24]) for a yaw-aligned actuator disk. Model predictions are shown using classical one-dimensional momentum modeling[7–9], Glauert's empirical relation[10], and the Unified Momentum Model introduced in this study. (Right) Schematic representation of the control volume enclosing the porous actuator disk that is used to derived the Unified Momentum Model developed in this study.

velocities, wake pressure, and power production. The unified equations predict the empirically observed monotonic increase in thrust coefficient with increasing induction factor, offering a first-principles solution to a persistent limitation of classical one-dimensional momentum modeling that predicts a trend in the opposite direction. The equations further predict the quantitative influence of rotor misalignment on wake velocities, thrust, and power at arbitrary, joint misalignments and thrust coefficients. As in classical momentum theory, the model is derived under uniform inflow. But relative to the classical theory, the Unified Model generalizes to predict the effects of both misalignment and high thrust operation without empirical corrections. The model therefore provides a new basis to account for additional effects such as turbulence and wind shear in the atmospheric boundary layer, rather than starting from classical theory coupled with empiricism. The Unified Momentum Model is coupled with blade element modeling to result in a new blade element momentum (BEM) modeling approach that can predict the effects of rotor misalignment and high thrust operation without empirical corrections in the momentum model. This study provides new theoretical insight and a simple, computationally-efficient model for complex turbine aerodynamics that have previously relied on empirical corrections or necessitated expensive CFD to resolve[12,13].

## Results

### Unified Momentum Model

In this study, we derive and validate an analytical relationship between the induction, streamwise and spanwise wake velocities, and the wake pressure through integral analysis of a control volume enclosing the actuator disk. This Unified Momentum Model addresses the two limiting assumptions of one-dimensionality and pressure recovery to freestream from classical momentum modeling. As in classical momentum theory, the analytical model is derived under uniform, inviscid inflow and neglects turbulence. As a result, the model is applicable at the rotor, predicting induction, thrust force, and power, and in the near-wake, which is the portion of the wake region before turbulent mixing contributes to the dynamics[44–47]. The induction is modeled using the Bernoulli equation, with the pressure drop at the actuator disk predicted by the thrust force. The radial flow is continuous over the porous actuator disk and therefore cancels in the Bernoulli equation, as further discussed in the Supplementary Information. The streamwise wake velocity is modeled using momentum conservation in the streamwise direction, along with mass conservation in the control volume and the streamtube enclosing the actuator disk. Both the induction and the streamwise wake velocity depend on each other and on the pressure deficit in the wake. Given an arbitrary misalignment angle between the actuator disk and the incident flow, the flow acquires a lateral component, deviating from the one-dimensional assumption of classical momentum theory. We model the lateral wake velocity using a lifting line model[26,48,49], that relates the lateral wake velocity to the lift component of the rotor-normal thrust force.

To close the system of equations, the remaining unknown is the pressure in the wake, which classical one-dimensional momentum modeling assumes to be equal to the freestream pressure[11]. As the induced velocity is increased, flow separation and a large pressure drop can occur[37,50], which prevents the wake pressure from recovering to its freestream value, and correspondingly increases the thrust[11]. To model this behavior, the wake pressure in this study is modeled using a solution to the steady, two-dimensional Euler equations with an actuator disk turbine body forcing, recently proposed by Madsen (2023)[51], based on the analytical method of von Kármán and Burgers (1935)[52]. The objective is to model the inviscid rotor and near-wake regions, therefore the inviscid Euler equations are used rather than the Reynolds-Averaged Navier–Stokes equations with a turbulence model, consistent with the assumption to neglect the effects of turbulence. In

the far-wake, turbulence[46] and three-dimensional effects such as wake curling[53] become important. The two-dimensional Euler equations are used in this study to result in an analytical model form, rather than requiring a CFD solver to predict a full three-dimensional pressure field. The quantitative fidelity of this modeling approximation is validated in the subsequent results.

The solution to the pressure Poisson equation is decomposed into contributions from the linear actuator disk body forces and the nonlinear advective terms. In the absence of the nonlinear advective pressure component, the Unified Momentum Model is entirely analytical. Model predictions with and without the nonlinear advective term are shown in the Supplementary Information, with the results demonstrating that including the nonlinear portion of the wake pressure lowers prediction error in the region of induction approximately between 0.7 and 0.9, and therefore, we determine that it is necessary to include. The nonlinear advective component of the pressure exhibits instability at very high thrust coefficients and induction factors (induction $\gtrsim 0.7$), in both the Euler equation solution and in large eddy simulations. We investigated mitigation solutions including convergence enhancement techniques and dynamic mode decomposition analysis, not shown for brevity. The final methodology selected records the minimum pressure at the centerline location throughout iterations of a range of relaxations, yielding an approximate upper bound for the magnitude of the nonlinear pressure drop. The full methodology is described in the Supplementary Information. The quantitative accuracy of this pressure modeling approach is evaluated in the results section, compared to large eddy simulations.

To complete this set of equations, a model for the near-wake length is needed to determine the pressure in the wake at the outlet of the inviscid near-wake region. The outlet of the inviscid near-wake region is coincident with the onset of the turbulent far-wake region, modeled using so-called wake models. Therefore, near-wake predictions from momentum theory are used to provide initial conditions to wake models[44–47]. Bastankhah & Porté-Agel (2016)[46] used the shear layer model of Lee & Chu (2003)[54] and one-dimensional momentum modeling to predict the near-wake length for low thrust and induction rotors. To predict the near-wake length, we adapt this approach to remain consistent with both high-thrust and yaw misaligned regimes simultaneously. The shear layer model introduces a single parameter, $\beta$, that relates the shear layer growth to the characteristic velocity shear between the wake and freestream velocity[46]. This parameter has a well-accepted range in the literature[54]. Here, the parameter is estimated using data from CFD of an actuator disk and the obtained value falls within the typical range documented in prior studies. Further, the quantitative influence of the parameter uncertainty is assessed in the results below, and in Supplementary Information. The wake expansion is also predicted by the model, consistent with a decrease in streamwise velocity and the lateral velocity, as connected through the conservation of mass and momentum.

The final system of equations, given in "Methods", provides an analytical relationship between the induction, streamwise velocity, lateral velocity, near-wake length, and wake pressure for a rotor with arbitrary yaw misalignments and thrust coefficients. The corresponding power production for a wind turbine or hydrokinetic rotor is computed using the predicted induction factor.

### Unified Momentum Model predictions in uniform flow

Predictions from the model proposed here are assessed through a priori and *a posteriori* analysis, compared to high-fidelity large eddy simulations (LES). To first focus on model assessment and validation, simulations are performed in the same dynamical context as the model development, namely, uniform inflow and an actuator disk modeled rotor. The numerical implementation of the LES is described in "Methods". To demonstrate the robustness of the numerical method,

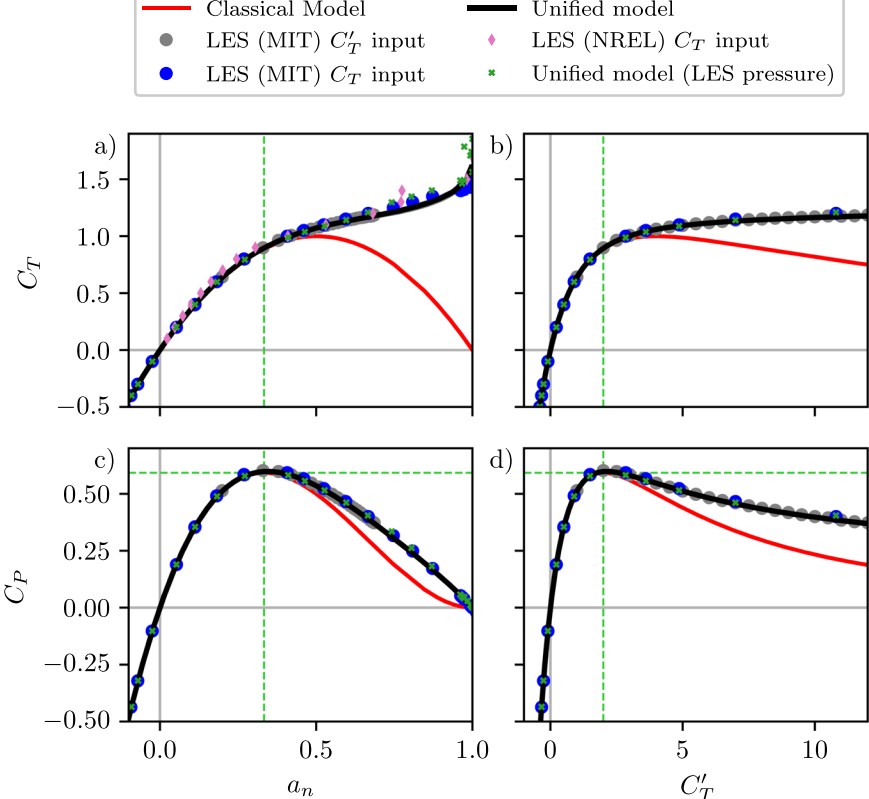

**Fig. 2 | Unified Momentum Model predictions of the coefficients of thrust and power compared to large eddy simulations.** Coefficient of thrust, $C_T$, as a function of (**a**) rotor-normal induction factor, $a_n$, and (**b**) the local thrust coefficient $C'_T$. Coefficient of power, $C_P$, as a function of (**c**) $a_n$, and (**d**) $C'_T$. The variables estimated by the presented Unified Momentum Model (Methods), are shown. The shaded region corresponds to $\pm 10\%$ uncertainty in parameter $\beta$. Note that this uncertainty range is visually negligible for the plotted quantities and therefore may not be visible. Results from large eddy simulation (LES) from MIT and NREL and classical one-dimensional momentum modeling are shown as a reference. A priori model results are also shown for reference, where the LES-measured pressure deficit $(p_1 - p_4)/(\rho u_\infty^2)$ is provided as an input to the model to facilitate prediction of $C_T$ and $C_P$. The Betz limit of $a_n = 1/3$ ($C'_T = 2$) and $C_P = 16/27$ is also shown by the dashed green line.

LES results are shown using two standard approaches for modeling the thrust force (see Calaf et al.[55]), depending on input of either the thrust coefficient $C_T$ and a thrust force dependent on the freestream velocity $\vec{u}_\infty$, or input of a modified thrust coefficient $C'_T$ and a thrust force dependent on the velocity at the rotor $\vec{u}_d$. Figure 2a illustrates the thrust coefficient $C_T$ as a function of the rotor-normal induction factor $a_n$ ($a_n = 1 - \vec{u}_d \cdot \hat{n}/\vec{u}_\infty \cdot \hat{n}$), which is the generalization of the classical one-dimensional induction factor. In addition to the present LES results, LES results from Martínez-Tossas et al. (2022, NREL)[13] are shown for reference. To demonstrate the validity of the model form, a priori model predictions are shown, where the wake pressure in LES is measured and provided to the model. The fully predictive *a posteriori* model output, where the pressure is modeled based on the methodology described previously, is also shown. The a priori model output exhibits remarkable agreement with LES in terms of predicting induction, wake velocities and power production, validating the model-form proposed here. The fully-predictive model, which requires no pressure input from LES, exhibits low error across the full range of rotor-normal induction factors realized in LES, showing substantial qualitative and quantitative predictive accuracy improvements compared to classical one-dimensional momentum modeling. The clear and notable deficiency of classical momentum modeling to predict the qualitative and quantitative response of the thrust coefficient $C_T$ for induction factors of $a_n \gtrsim 1/3$ has resulted in numerous empirical formulas to be proposed in the literature[14,15,21,40]. In this study, through first-principles modeling of the wake pressure, substantial increases in predictive accuracy are achieved without empirical formulas. Similar observations can be made through analysis of the coefficient of power

$C_P$ depending on the rotor-normal induction factor $a_n$ as shown in Fig. 2c.

Interestingly, the proposed model predicts the maximum value of $C_P$ as 0.5984, occurring at an induction factor of 0.345, which is ~1.0% and 3.5% higher than the classical Betz limit[38] (also known as the Lanchester-Betz-Joukowsky limit[23,56,57]) for the coefficient of power (16/27) and power-maximizing induction factor (1/3), respectively. Since the Betz limit is derived using classical one-dimensional momentum modeling with the assumption that the wake pressure recovers to freestream conditions, it inherently dictates that the turbine cannot extract energy from the static pressure. In contrast, the model presented here does account for the persistent pressure drop in the far wake, resulting in a marginal increase in energy extraction. While these are modest departures from the classical Betz limit, this result elucidates that neglecting the energetic contributions of the wake pressure deficit will incorrectly model energy conservation. This result is further highlighted by the consistent underprediction of the coefficient of power $C_P$ of one-dimensional momentum modeling for induction factors of $a_n \gtrsim 1/3$. The consistent one-dimensional momentum modeling error in the coefficient of power for $a_n \gtrsim 1/3$ is alleviated by the pressure model proposed here. In Fig. 2b, and d, the coefficients of thrust and power are shown as a function of the modified thrust coefficient $C'_T$[55] for $0 < C'_T < 12$, which is a common alternative method to model actuator disks in CFD. As is evident in Fig. 2a, the marginal increase in $C_T$ is reduced for higher values of $C'_T$, such that $C'_T = 12$ results in an induction factor of only $a_n \approx 0.7$. This results from the definition of the thrust described previously, where the thrust force depends on the local thrust coefficient $C'_T$ and the disk velocity $\vec{u}_d$. As

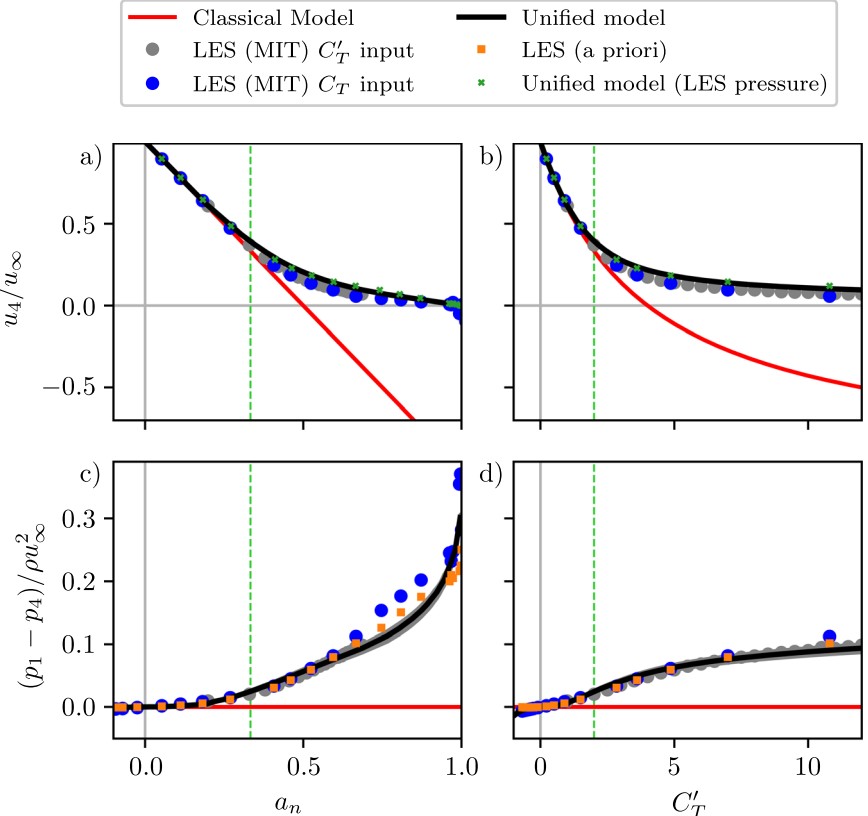

**Fig. 3 | Unified Momentum Model predictions of wake velocity and wake pressure compared to large eddy simulations.** Wake streamwise velocity $u_4$ as a function of (**a**) rotor-normal induction factor, $a_n$, and (**b**) local thrust coefficient $C_T'$. Density-normalized wake pressure deficit $(p_1 - p_4)/(\rho u_\infty^2)$ as a function of (**c**) $a_n$ and (**d**) $C_T'$. The variables estimated by the presented Unified Momentum Model (Methods) are shown. The shaded region corresponds to $\pm 10\%$ uncertainty in $\beta$. Note that this uncertainty range is visually negligible for some plotted quantities and therefore may not be visible. Results from LES and classical one-dimensional momentum modeling are shown as a reference. Two additional a priori model results are shown. For the wake streamwise velocity $u_4$ (**a, b**), a priori model results are shown where the LES measured pressure deficit $(p_1 - p_4)/(\rho u_\infty^2)$ is provided as an input to the model to facilitate prediction of wake velocity $u_4$. For the wake pressure deficit $(p_1 - p_4)/(\rho u_\infty^2)$ (**c, d**), the pressure deficits that are implied from energy conservation, with the input of LES measured $a_n$ and $u_4$, are shown. The Betz limit of $a_n = 1/3$ and $C_T' = 2$ is also shown by the dashed green line.

the local thrust coefficient $C_T'$ is increased, the induction also increases. Correspondingly, the disk velocity $\vec{u}_d$ is lowered, which in turn lowers the thrust force. The counteracting effects of increasing $C_T'$ and decreasing disk velocity limit the growth in the thrust coefficient $C_T$, which depends on these variables jointly. Since thrust force and power produced by rotors are based on the velocity at the rotor ($\vec{u}_d$), the local thrust coefficient model offers a more physically consistent representation of a wind turbine rotor[49,55].

Since the modeling framework proposed here represents the relationship between thrust, power, and wake variables from first-principles, the predicted wake velocities conserve mass, momentum, and energy across arbitrary thrust coefficients and misalignment angles. The streamwise wake velocity and the density-normalized wake pressure deficit are shown in Fig. 3a and c, respectively, as a function of the rotor-normal induction $a_n$, and in Fig. 3b,d as a function of the modified thrust coefficient $C_T'$. The methodology for computing wake quantities $u_4$, $v_4$, $x_0$, and wake pressure from LES is described in the "Streamtube analysis numerical setup" in "Methods". Above $a_n = 0.5$ and $C_T' = 4$, classical momentum modeling predicts that the wake velocity is negative, which differs from the LES output. Instead, both the LES and the model predictions (a priori and *a posteriori*) asymptote to zero wake velocity for $a_n = 1$ ($C_T' \rightarrow \infty$).

The density-normalized wake pressure deficit (Fig. 3c) from LES is first compared to the wake pressure that is implied by energy conservation in the control volume (further described in Supplementary Information). To estimate the wake pressure implied by energy

conservation, we solve for the wake pressure deficit using the model equation for the induction factor $a_n$, and we input the LES measured values for $a_n$ and the wake velocities. As shown in Fig. 3c, there is excellent agreement between the LES measured wake pressure and the wake pressure implied by energy conservation for $a_n < 0.7$, further validating the model form. As $a_n$ approaches unity, the accuracy of the Bernoulli equation in the wake is reduced, which lowers the quantitative accuracy of the pressure prediction in this region, but this has a limited quantitative impact on the primary variables of interest ($C_T$, $C_P$, and wake velocities shown in Fig. 2 and Fig. 3a). The reduction in the accuracy of the Bernoulli equation is related to the increasing degree of turbulence in the region immediately downwind of the turbine. The length of the near-wake reduces as the thrust coefficient increases (further detailed in Supplementary Information). This trend is captured in the model. However, even with a shrinking near-wake length, neglecting turbulent mixing incurs increasing error with increasing thrust. Still, the Bernoulli equation is used in the model form to preserve the computational efficiency of the analytical formulation. Finally, we show the fully predictive wake pressure results from the differential Euler equations and near-wake length closure proposed here. The predictive pressure model exhibits qualitative and quantitative agreement, with some increasing predictive error at values of induction above $a_n > 0.7$ from the Bernoulli equation as described previously. As the induction $a_n$ approaches unity, there is a nonlinear increase in the wake pressure deficit, with the limiting state of $a_n = 1$ resulting in flow separation[37,50]. At and above an induction of unity

$(a_n \geq 1)$, the wake flow is separated, resulting in the bluff body wake of a flat circular plate. In such cases, the presented model is no longer valid. While the regime of unity induction $(a_n \geq 1)$ is likely not often relevant to wind power applications, it is useful to connect this Unified Momentum Model with research focused on bluff body wake dynamics[58,59].

A parallel modeling approach to the integral analysis presented here represents the porous disk with a distribution of equal magnitude sources in potential flow[60–62]. However, these approaches have also in large part neglected the wake pressure deficit, similar to classical one-dimensional momentum modeling. Steiros & Hultmark (2018)[50] extended the source modeling approach by providing a more detailed representation of the wake, using momentum conservation and the Bernoulli equation, and by including a wake pressure term. To close the derived system of equations analytically without an additional pressure model equation, a simple wake factor was introduced to model the streamwise wake velocity. The one-dimensional model proposed by Steiros & Hultmark (2018)[50] exhibited excellent agreement compared to experimental measurements of the coefficient of drag (thrust) of porous plates immersed in a water tank. While the thrust predictions of this model exhibit substantial improvements for high values of induction compared to classical one-dimensional momentum modeling[50], it is notable that the wake factor-based wake velocity model differs from classical one-dimensional momentum model predictions and CFD data, even for low induction values (below the heavily loaded limit of $a_n \approx 0.37$[25]). Detailed comparisons between the model proposed here and the Steiros & Hultmark (2018)[50] model are made in the Supplementary Information.

We further evaluate the model predictions for yaw misaligned rotors. Accurately predicting the power output of a yaw-misaligned wind turbine is crucial, given that turbines typically operate in yaw[27], and for the effective flow control in wind farms through wake steering[5,36]. Recently, Heck et al. (2023)[49] demonstrated that the deviation of yawed rotor power production from $\cos^3(\gamma)$ results from the impact of the yaw misalignment on the induction factor $a_n$, but the proposed model had increasing error with increasing thrust coefficients because it assumed that the wake pressure recovers to freestream pressure. The presented Unified Momentum Model extends the framework introduced by Heck et al. (2023), ensuring its validity at high thrust coefficients by relaxing the pressure recovery assumption. As previously discussed, the one unknown parameter $\beta$ unspecified in the Unified Momentum Model was calibrated using LES data of yaw-aligned turbines to find that it was well within the range accepted in literature. Here, we use that fixed value of $\beta$ calibrated to yaw-aligned LES data to predict out-of-sample conditions for a yaw misaligned turbine. In the Supplementary Information, we show predictions for the near-wake length as a function of the yaw misalignment without re-calibrating $\beta$, demonstrating confidence in both the value of the parameter and its ability to accurately generalize to unseen conditions. In Fig. 4, the coefficient of power $C_P$ is shown for yaw misalignments between $0 < \gamma < 50°$ and local thrust coefficients between $0 < C_T' < 4$ for the proposed model and for LES, along with a baseline model from classical one-dimensional momentum modeling. The model proposed here exhibits high levels of qualitative and quantitative accuracy compared to LES. Specifically, the model demonstrates an 84% and a 21% reduction in the mean absolute error of $C_P$ across the thrust coefficient and yaw misalignment values considered here compared to classical one-dimensional momentum modeling (with $P(\gamma) = P(\gamma = 0) \cdot \cos^3(\gamma)$) and to the model that neglects the wake pressure deficit[49], respectively.

## A Blade Element Momentum (BEM) model based on the Unified Momentum Model

The Unified Momentum Model generalizes and directly replaces classical one-dimensional momentum theory in its many uses. We note that in the limit of neglecting the wake pressure and yaw misalignment, the Unified Momentum Model yields identical predictions to the limited classical theory. Two important applications for inviscid momentum modeling of rotors are to provide initial conditions to far-wake models that account for turbulence[44–47] and to provide the necessary closure for blade element momentum (BEM) modeling for rotors[14,15,63]. In state-of-the-art and widely-used BEM modeling, classical momentum theory deficiencies in high thrust and yaw misalignment are universally handled with empirical corrections[14,15].

We develop a first-principles blade element momentum model based on the Unified Momentum Model that predicts high thrust and yawed operation without empirical corrections for these states for the first time. The classical momentum theory conventionally employed alongside numerous empirical corrections for high-thrust and yaw-misaligned scenarios is directly replaced in the BEM model with the Unified Momentum Model. There is no need to modify the blade element portion of BEM. The advantage of BEM modeling over momentum modeling alone is that BEM models realistic turbine controller parameters, such as blade pitch angle and rotor angular velocity (typically nondimensionalized as a tip-speed ratio) and their effect on rotor aerodynamics and forces (loads). Using the unified BEM model, the coefficient of power over a range of blade pitch angles and rotor tip-speed ratios is predicted and shown in Fig. 5. We compare the unified BEM model developed here to standard approaches currently used. Using a BEM model dependent on classical one-dimensional momentum theory without a high-thrust correction (Fig. 5a) fails to converge in regions where the thrust coefficient exceeds unity, rendering its solution undefined. Consequently, the global optimal operating point using this classical model lies on the convergence boundary, preventing the accurate determination of the optimal controller set point. By incorporating an empirical high-thrust correction (Fig. 5b), the upper-left quadrant of the coefficient of power surface can be realized, allowing a globally optimal set point to be retrieved, but it requires empiricism to achieve which yields uncertainty. Replacing the classical momentum theory and empirical high-thrust correction with the Unified Momentum Model (Fig. 5c) enables the prediction of the coefficient of power across all of the relevant pitch and tip-speed ratio regimes, and therefore the identification of the optimal control strategy without any empirical corrections.

The BEM models are used to identify the pitch and tip-speed ratio that maximizes the coefficient of power for the turbine. When the turbine is yaw misaligned, classical momentum theory without or with corrections (Fig. 5d, e, respectively) predicts a decrease in power maximizing tip-speed ratio. This tip-speed ratio reduction causes a reduction in thrust force. In contrast, the BEM implementation using the Unified Momentum Model (Fig. 5f) predicts that the power maximizing control strategy entails a decrease in pitch angle, which increases the thrust level, relative to fixing control at the yaw-aligned power maximizing set-point. This result agrees with recent literature which indicates that optimal control of wind turbines with yaw misalignment requires an increase in the thrust coefficient[32,64]. The novel BEM model based on the Unified Momentum Model is compared to blade-resolved wind turbine simulations[65], demonstrating a threefold reduction in error for yaw of 30° compared to classical momentum theory. Further results comparing the BEM based on the Unified Momentum Model to standard empirical approaches are provided in the Supplementary Information. In summary, the Unified Momentum Model yields a new approach to blade element momentum modeling that predicts yawed and high thrust operation from first-principles without empirical corrections in the momentum model. The new first-principles model also enables new engineering insights, for example, identifying the control strategy for a yaw misaligned turbine to maximize power production. This recommended control strategy qualitatively differs from the control recommendations predicted by classical momentum theory.

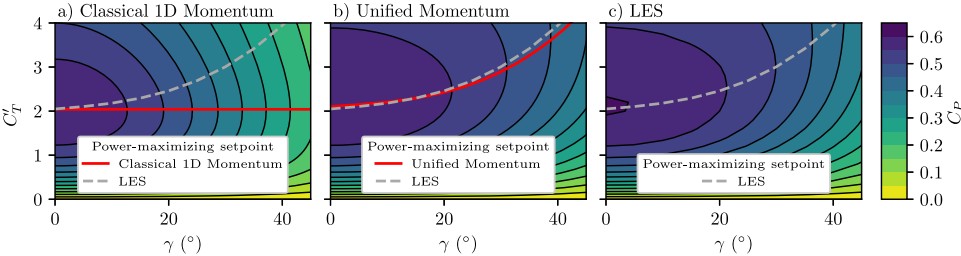

**Fig. 4 | Coefficient of power depending on thrust coefficient and yaw mis-alignment for the Unified Momentum Model, large eddy simulation, and classical momentum modeling.** Coefficient of power $C_P$ as a function of the yaw misalignment $\gamma$ and the local thrust coefficient $C_T'$ for (**a**) classical one-dimensional momentum modeling and empirically modeling the effect of yaw misalignment as $P(\gamma) = P(\gamma = 0) \cdot \cos^3(\gamma)$ which is a common model[11] given the previous lack of a first-principles approach, (**b**) the presented Unified Momentum Model (Methods), and (**c**) LES. The power-maximizing $C_T'$ set points as a function of $\gamma$ are indicated in red.

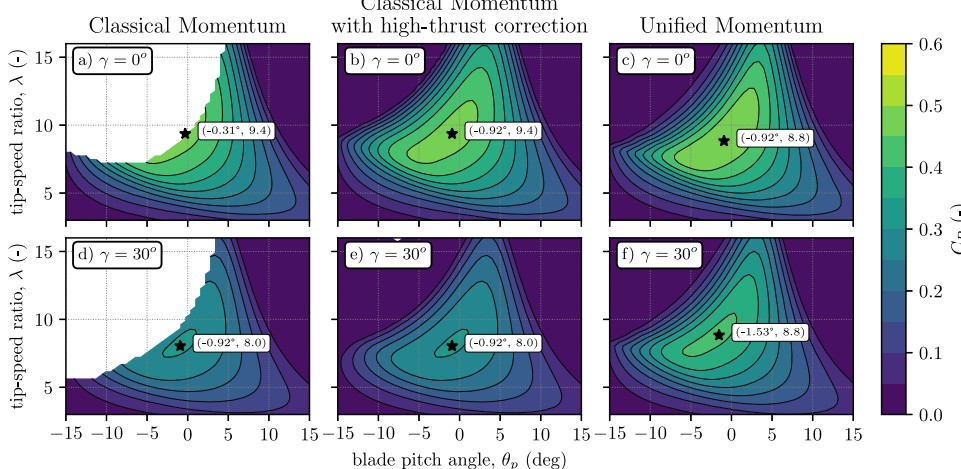

**Fig. 5 | Blade element momentum modeling based on the Unified Momentum Model.** Contour plots showing the variation of power coefficient ($C_P$) with blade pitch angle ($\theta_p$) and blade tip-speed ratio ($\lambda$) using different thrust-induction momentum modeling closures in a blade element momentum (BEM) model implementation. Fully aligned (**a**–**c**) and yaw misaligned (**d**–**f**) conditions at $\gamma = 30°$ are considered. The momentum modeling closure used in the BEM includes (**a** and **d**) classical momentum theory, (**b** and **e**) classical momentum with high-thrust correction, and (**c** and **f**) the Unified Momentum Model, all incorporating Prandtl tip and root correction[83] in the blade element model. $C_P$-maximizing set-points are indicated with a marker.

## Unified Momentum Model predictions in atmospheric boundary layer flow

As in one-dimensional momentum theory, the Unified Momentum Model developed here is derived in the context of turbulence-free, uniform inflow across the disk area. However, one-dimensional momentum theory is ubiquitously used for predictions in realistic environments with turbulence and wind shear in BEM and wake modeling applications. Similarly, we investigate the fidelity of the Unified Momentum Model in turbulent atmospheric boundary layer (ABL) conditions typical for these wind power applications. First, in Fig. 6a, c, e, and g, comparisons between the Unified Momentum Model and uniform inflow LES are shown for rotor quantities $a_n$ and $C_T$, as well as near wake velocities in the streamwise $\delta u_0 = u_\infty - u_4$ and spanwise $\delta v_0 = v_\infty - v_4$ directions, where the $\infty$ subscript denotes the inflow velocities. Over a wide range of yaw angles $\gamma \in [0, 40°]$ and local thrust coefficients $C_T' \in [0.4, 9.6]$, the Unified Model quantitatively predicts relevant near-wake properties with high accuracy. Second, we investigate the predictions of these same quantities for an actuator disk immersed in ABL conditions using a comprehensive suite of large eddy simulations over a range of thrust coefficients and yaw angles. The numerical setup used for the seventy independent LES runs of the ABL is provided in "Methods". We show a comparison between near-wake quantities extracted from LES with turbulent ABL inflow and the Unified Momentum Model in Fig. 6b, d, f, and h for a range of yaw

misalignment angles $\gamma_1 \in [0, 45°]$ and thrust coefficients $C_T' \in [0.4, 4]$. In general, model errors are greater in ABL inflow than for uniform inflow, particularly at the coincidence of high yaw-misalignment angles and thrust coefficients, as expected based on the dynamical regime considered in the model derivation. Still, the Unified Momentum Model lowers prediction error in turbulent ABL inflow across these yaw and thrust coefficient regimes by 60%, 83%, and 78% for the induction, streamwise wake velocity, and spanwise wake velocity, respectively, compared to classical one-dimensional momentum theory.

## Discussion

The simple, first-principles analytical relationship between the induction, thrust, power, wake velocities, near-wake length, and wake pressure proposed here is ideally suited for implementation in a wide range of applications, such as blade-element momentum (BEM) modeling frameworks for wind and hydrokinetic turbines and propellers, such as OpenFAST[42] and HAWC2[66], as well as in wake models used for wind farm design and control, such as the FLORIS model[41] and PyWake[67]. The predictive Unified Momentum Model developed here, which has a runtime on the order of microseconds on a standard desktop computer, has lower predictive error than classical one-dimensional momentum modeling for all thrust coefficients and misalignment angles considered, enabling rotor modeling for arbitrary thrust and yaw conditions without empirical corrections for the first

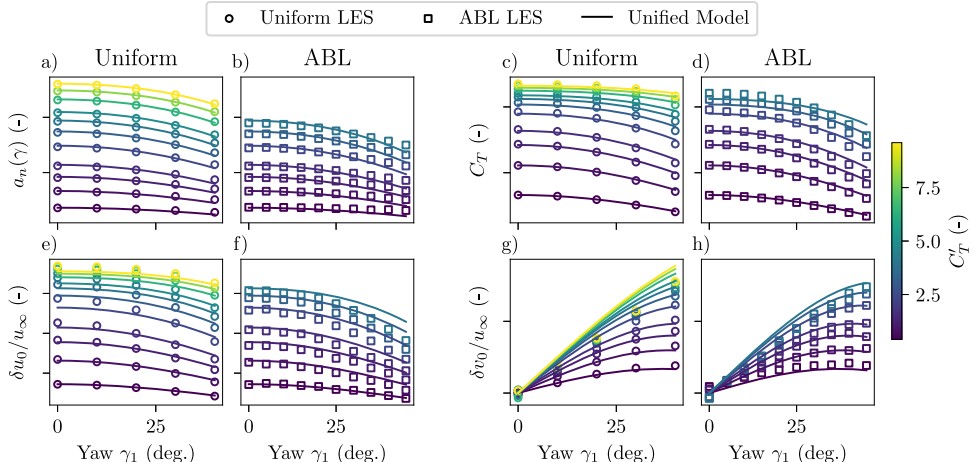

**Fig. 6 | Unified Momentum Model predictions for wind turbines in the atmospheric boundary layer.** Model comparison for rotor and near-wake properties, measured as streamtube-averaged quantities, in uniform and atmospheric boundary layer (ABL) inflow as a function of local thrust coefficient $C'_T$ and yaw $\gamma$. Subfigures show (**a**, **b**) rotor-normal induction factor, (**c**, **d**) thrust coefficient $C_T$, (**e**, **f**) initial streamwise velocity deficit $\delta u_0 = u_\infty - u_4$, and (**g**, **h**) initial lateral velocity deficit $\delta v_0 = v_\infty - v_4$. Model predictions are compared with LES in (**a**, **c**, **e**, **g**) uniform inflow and (**b**, **d**, **f**, **h**) conventionally neutral ABL conditions.

time. The first-principles BEM model developed here based on the Unified Momentum Model accurately predicts the power and forces of a turbine without empirical corrections tailored to the high thrust and yaw-misaligned states, yielding new insights into the design and optimization of wind turbine control. The Unified Momentum Model improves our physical understanding of rotors operating across all thrust regimes, including a modification to the Betz limit[23,38] that is enabled by momentum and kinetic energy extraction from the pressure. Future work is encouraged to enhance the understanding of the shear layer growth parameter, $\beta$. Although this parameter exhibits a weak sensitivity on rotor quantities such as thrust and power, it exerts a more significant impact on wake variables, particularly governing the balance between wake pressure and wake velocity (see Supplementary Information). Further investigation is recommended to explore the universality of this parameter in comparison to high Reynolds number experimental measurements.

The growing trends in wind turbine design, such as larger rotor diameters, taller hub heights, and increasing complexity in design and control strategies, are pushing the limits of the applicability of existing modeling tools. Consequently, the scientific community is motivated to address grand scientific challenges[4], focused on the design[68] and control[5] of modern utility-scale wind turbines and wind farms. The proposed Unified Momentum Model is a first-principles-based approach that unifies rotor thrust, yaw, and induction with the outlet velocity and pressure of flow through an actuator disk. It addresses major theoretical limitations in modeling rotor performance under high thrust and yaw misalignment. These complex operating conditions have historically relied on empirical models due to gaps in underlying theory. Such empirical approaches have been deployed widely across wind power applications, affecting BEM modeling employed for wind turbine design and control, as well as the wake models utilized for wind farm design and operation. Derived under assumptions of uniform, zero-turbulence inflow, the Unified Momentum Model provides an ideal starting point for extensions to more complex operational regimes such as turbulent, non-uniform inflow and rotational effects, rather than starting from a basis that already includes empirical models that are unlikely to extrapolate out-of-sample. The Unified Momentum Model yields equal or lower predictive error compared to LES than classical one-dimensional momentum theory for the yaw angles and thrust coefficients investigated here. Low predictive error is exhibited by the presented results across uniform inflow regimes and most atmospheric cases, yet

opportunities persist to enhance modeling accuracy for high thrust coefficients and yaw angles when accounting for ABL flows. Future work should extend the analysis and model to consider unsteadiness, such as from floating motion for an offshore turbine, wind speed and direction shear, and turbulence.

## Methods

### Unified Momentum Model

This study employs integral analysis of the control volume enclosing the actuator disk (Fig. 1) to derive a system of analytical equations describing the relationships between rotor-normal induction, thrust force, power production, wake pressure, near-wake length, and wake outlet velocities. The derivation is described in detail in the model derivation section of the Supplementary Information. The analytical equations consider both yaw-aligned and yaw-misaligned (porous) actuator disks. The inputs are the modified thrust coefficient $C'_T$ and yaw misalignment angle $\gamma$. The modified thrust coefficient controls the thrust force as $\vec{F}_T = -\frac{1}{2}\rho C'_T A_d (\vec{u}_d \cdot \hat{n})^2 \hat{n}$, where $\vec{u}_d$ is the velocity at the actuator disk, $\rho$ is the fluid density, $A_d$ is the disk area, and $\hat{n}$ is the unit vector normal to the disk. The thrust force can be written as a function of the rotor-normal induction factor $a_n$ and the yaw angle $\gamma$, $\vec{F}_T = -\frac{1}{2}\rho C'_T A_d (1 - a_n)^2 \cos^2(\gamma) u_\infty^2 [\cos(\gamma)\hat{i} + \sin(\gamma)\hat{j}]$. The equations then solve for the rotor-normal induction $a_n$, streamwise wake velocity $u_4$, lateral wake velocity $v_4$, near-wake length $x_0$, and the pressure difference $(p_4 - p_1)$ as outputs. The final form of the equations is:

$$a_n = 1 - \sqrt{\frac{u_\infty^2 - u_4^2 - v_4^2}{C'_T \cos^2(\gamma) u_\infty^2} - \frac{(p_4 - p_1)}{\frac{1}{2}\rho C'_T \cos^2(\gamma) u_\infty^2}} \tag{1}$$

$$
\begin{aligned}
u_4 = &-\frac{1}{4} C'_T (1 - a_n) \cos^2(\gamma) u_\infty \\
&+ \frac{u_\infty}{2} + \frac{1}{2}\sqrt{\left(\frac{1}{2} C'_T (1 - a_n) \cos^2(\gamma) u_\infty - u_\infty\right)^2 - \frac{4(p_4 - p_1)}{\rho}}
\end{aligned} \tag{2}
$$

$$v_4 = -\frac{1}{4} C'_T (1 - a_n)^2 \sin(\gamma) \cos^2(\gamma) u_\infty \tag{3}$$

$$\frac{x_0}{D} = \frac{\cos(\gamma)}{2\beta} \frac{u_\infty + u_4}{|u_\infty - u_4|} \sqrt{\frac{(1-a_n)\cos(\gamma)u_\infty}{u_\infty + u_4}} \tag{4}$$

$$p_4 - p_1 = -\frac{1}{2\pi}\rho C_T'(1-a_n)^2\cos^2(\gamma)u_\infty^2 \arctan\left[\frac{1}{2}\frac{D}{x_0}\right] + p^{NL}(C_T',\gamma,a_n,x_0), \tag{5}$$

where the freestream incident wind speed is $u_\infty$, the actuator disk diameter is $D$, and the unknown shear layer growth rate parameter is $\beta = 0.1403$ as outlined in the Supplementary Information. The pressure equation (Eq. (5)) contains two terms. The first term is the pressure contribution from the actuator disk forcing, and the second term ($p^{NL}$) is a nonlinear term that results from the advection. The variable $p^{NL}$ is the nonlinear pressure contribution to the outlet wake pressure, which is described in detail in the model derivation in Supplementary Information.

In some applications, utilizing the thrust coefficient $C_T$ as the input parameter proves to be convenient compared to using the local thrust coefficient $C_T'$, for instance, when performing blade-element modeling of a rotor where the thrust coefficient is output by the blade-element model to be input into the momentum model. Another circumstance is when a utility-scale wind turbine's thrust curve is available as a function of freestream wind speed $u_\infty$, adopting $C_T$ as the input variable can offer convenience. Because $C_T$ is a derived quantity that depends on both the induction and the yaw misalignment angle, to use $C_T$ as an input variable requires a sixth equation to be included in the set of equations above:

$$C_T' = \frac{C_T}{(1-a_n)^2\cos^2(\gamma)}, \tag{6}$$

where the thrust coefficient is defined as $C_T = 2\|\vec{F}_T\| / (\rho A_d u_\infty^2)$, where $\|\vec{F}_T\|$ is the magnitude of the thrust force, which is further defined in the following section. Although it is possible to algebraically reformulate Eq. (1)–(5) in different ways to utilize the thrust coefficient $C_T$ as an input parameter, the presented set of six equations offers numerically robust solutions applicable to a broad range of input values and initial conditions when solved using fixed-point iteration as described in the Supplementary Information. We emphasize that the two forms of the model, whether $C_T'$ or $C_T$ is input, are mathematically and physically exactly equivalent. The only difference is how one chooses to represent the thrust force $\vec{F}_T$ in notation, but it is important that the thrust coefficient is defined to be consistent with the model form and geometry presented in this study.

## Large eddy simulation numerical setup

Large eddy simulations are performed using PadéOps[69,70], an open-source incompressible flow code[71]. The horizontal directions use Fourier collocation and a sixth-order staggered compact finite difference scheme is used in the vertical direction[72]. A fourth-order strong stability preserving (SSP) variant of the Runge-Kutta scheme is used for time advancement[73] and the sigma subfilter scale model is used[74]. Simulations are performed with uniform inflow with zero freestream turbulence, consistent with the derivation of classical momentum modeling. The boundary conditions are periodic in the $x$ and $y$ directions with a fringe region[75] used in the $x$-direction to remove the wake from recirculating. The simulations are performed with a domain size of $L_x = 25D$ in length and cross-sectional size $L_y = 20D$, $L_z = 10D$ with $256 \times 512 \times 256$ grid points and with the turbine $5D$ downwind of the inlet.

Additional simulations with atmospheric boundary layer (ABL) inflow conditions are run using the concurrent-precursor method[76]. An empty, horizontally homogeneous domain without turbines is used to spin up turbulence using the initialization methodology from Liu

et al.[77] with a surface roughness $z_0 = 1\,\text{mm}$, a driving geostrophic wind speed of $G = 8\,\text{m s}^{-1}$, and a free atmosphere lapse rate of $\Gamma = 1\,\text{K km}^{-1}$. The Coriolis parameter $f_c$ is set to $10^{-4}\,\text{rad s}^{-1}$. The ABL LES domain is $3.84\,\text{km} \times 1.28\,\text{km} \times 1.28\,\text{km}$ in the streamwise, lateral, and vertical directions, with a grid spacing of $\Delta x = 10\,\text{m}$, $\Delta y = 5\,\text{m}$, and $\Delta z = 5\,\text{m}$, respectively. Periodic boundary conditions are used in the lateral direction, and the bottom wall uses a wall model based on Monin-Obukhov similarity theory[70,78]. A single actuator disk with a diameter of 100 m is placed $5D$ from the inlet at a hub height of 100 m in the ABL LES simulations.

The porous disk is modeled using an actuator disk model (ADM), that imparts a thrust force that depends on the modified thrust coefficient $C_T'$ and the disk velocity $\vec{u}_d \cdot \hat{n}$[55]

$$\vec{F}_T = -\frac{1}{2}\rho C_T' A_d(\vec{u}_d \cdot \hat{n})^2 \hat{n}, \tag{7}$$

where $\rho$ is the fluid density, $A_d = \pi D^2/4$ is the area of the disk where $D$ is the diameter, and $\hat{n}$ is the unit normal vector perpendicular to the disk. Note that this differs from the simplified model of thrust force from an actuator disk that depends on the freestream rotor-normal wind speed $\vec{u}_\infty$, $\vec{F}_{T,\text{ideal}} = -\frac{1}{2}\rho C_T A_d \|\vec{u}_\infty\|^2\hat{n}$, where $\vec{u}_\infty = u_\infty\hat{i} + 0\hat{j}$ is the freestream wind velocity vector and $C_T = 2\|\vec{F}_{T,\text{ideal}}\| / (\rho A_d \|\vec{u}_\infty\|^2)$. Porous disks produce thrust based on the wind velocity at the rotor, which has been modified by induction. The coefficient of thrust $C_T$ is an empirical quantity that depends on the magnitude of the induction, and it needs to be measured or predicted using a model. It is preferable for both analytical and numerical modeling, and more physically intuitive, to model the thrust force based on the velocity that is accessible to the porous disk, which is the disk velocity $\vec{u}_d$, and thus $C_T'$ is the input thrust coefficient. For a yaw-aligned turbine, $C_T' = C_T/(1-a(\gamma=0))^2$[55].

The rotor-normal, rotor-averaged induction factor $a_n$ for a disk with yaw misalignment angle $\gamma$ is defined as

$$a_n = 1 - \frac{\vec{u}_d \cdot \hat{n}}{u_\infty \cos(\gamma)}. \tag{8}$$

In the yaw-aligned case, the rotor-normal induction factor reduces to the standard axial induction factor $a = 1 - u_d/u_\infty$. Combining Eqs. (7) and (8), the thrust force written in terms of the rotor-normal induction factor is then

$$\vec{F}_T = -\frac{1}{2}\rho C_T' A_d(1-a_n)^2\cos^2(\gamma)u_\infty^2[\cos(\gamma)\hat{i} + \sin(\gamma)\hat{j}]. \tag{9}$$

The power for an actuator disk-modeled wind turbine is computed as $P = -\vec{F}_T \cdot \vec{u}_d$.

The numerical ADM implementation follows the regularization methodology introduced by Calaf et al.[55] and further developed by Shapiro et al.[79]. The porous disk thrust force $\vec{f}(\vec{x}) = \vec{F}_T \mathcal{R}(\vec{x})$ is implemented in the domain ($\vec{x}$) through an indicator function $\mathcal{R}(\vec{x})$. The indicator function $\mathcal{R}(\vec{x})$ is $\mathcal{R}(\vec{x}) = \mathcal{R}_1(x)\mathcal{R}_2(y,z)$ where

$$\mathcal{R}_1(x) = \frac{1}{2s}\left[\text{erf}\left(\frac{\sqrt{6}}{\Delta}\left(x+\frac{s}{2}\right)\right) - \text{erf}\left(\frac{\sqrt{6}}{\Delta}\left(x-\frac{s}{2}\right)\right)\right], \tag{10}$$

$$\mathcal{R}_2(y,z) = \frac{4}{\pi D^2}\frac{6}{\pi\Delta^2}\iint H\left(D/2 - \sqrt{y'^2+z'^2}\right)\exp\left(-6\frac{(y-y')^2+(z-z')^2}{\Delta^2}\right)\text{d}y'\,\text{d}z', \tag{11}$$

where $H(x)$ is the Heaviside function, $\text{erf}(x)$ is the error function, $s = 3\Delta x/2$ is the ADM disk thickness, and $\Delta$ is the ADM filter width. The disk velocity $\vec{u}_d$, used in the thrust force calculation Eq. (7), is calculated using the indicator function, $\vec{u}_d = M\mathcal{R}(\vec{x})\vec{u}(\vec{x})\,\text{d}^3\vec{x}$, where

$\bar{u}(\vec{x})$ is the filtered velocity in the LES domain and $M$ is a correction factor that depends on the filter width $\Delta$[79]. Small values of filter width $\Delta$ tend towards the theoretical actuator disk representation (a true actuator disk represents the disk as infinitesimally thin with a discontinuity in forcing at radial position $r = R$), but lead to numerical oscillations in LES. Given the nature of spatially distributing the thrust force with a larger value of $\Delta$, numerical implementations of the ADM typically underestimate the induction and therefore overestimate power production[79,80]. The correction factor $M = (1 + (C_T'\Delta/(2\sqrt{3\pi}D)))^{-1}$ derived by Shapiro et al.[79] is used to correct this error by ensuring that the filtered ADM in Eq. (10) sheds the same amount of vorticity as the infinitesimally thin disk, depending on $C_T'$ and the ADM filter width. Here, we use $\Delta/D = 3h/(2D)$ where $h = (\Delta x^2 + \Delta y^2 + \Delta z^2)^{1/2}$ is the effective grid spacing. The qualitative and quantitative conclusions of this paper are not affected by this choice, as shown in "Comparison between different actuator disk model regularization methods in LES" in the Supplementary Information, provided that the correction factor $M$ derived by Shapiro et al.[79] is used for larger $\Delta$. For small $\Delta$ values (e.g. $\Delta/D = 0.032$), numerical (grid-to-grid) oscillations contaminate the wake pressure measurements. Therefore, we have selected $\Delta/D = 3h/(2D)$ with $M$ given above. In summary, sensitivity experiments are performed for different numerical ADM implementations in Supplementary Information, and the results demonstrate that the qualitative conclusions of this study do not depend on the numerical implementation of the ADM.

### Streamtube analysis numerical setup

We consider a three-dimensional streamtube analysis. While a yaw-aligned actuator disk in uniform inflow presents an axisymmetric streamtube, yaw misalignment results in wake curling[53] and three-dimensional variations that motivate a three dimensional streamtube analysis. A theoretical actuator disk model has uniform thrust force for all positions $r < R$ within the rotor, where $r$ is a radial position defined relative to the disk center and $R$ is the radius of the disk. Numerical implementations of actuator disk models for computational fluid dynamics use regularization methodologies to avoid sharp discontinuities in the wind turbine body force[49,55,79], as discussed in "Large eddy simulation numerical setup", which results in some variation in the thrust force towards the outer extent of the disk radius. The degree to which there is variation in the thrust force for $r < R$ depends on the numerical implementation of the regularization (i.e. filter length $\Delta$ in the present implementation). Following previous studies[26,49], to focus the streamtube analysis on the portion of the wake over which the thrust force is constant, the streamtube seed points are defined at the actuator disk with an initial radius of $R_s < R$, where $R_s/R = 0.7$. Numerical tests (not shown) demonstrate a small quantitative sensitivity between $0.5 < R_s/R < 0.9$, but the qualitative results are insensitive to $R_s$.

While flow quantities within the streamtube depend on $x$, a core assumption in near-wake models is that there is a particular $x$ location (or a range of $x$ locations), that is characteristic of the inviscid near-wake (potential core), such that near-wake flow quantities can be described with a single value per variable, rather than a one-dimensional field variable depending on $x$. The fidelity of this assumption is investigated in "Streamtube analysis and budgets" in Supplementary Information. Here, we describe the methodology to define these individual flow quantities from LES data. The near-wake streamwise velocity $u_4$ is taken as the minimum value of the streamtube-averaged streamwise velocity on the interval $0 < x/D < 5$. This is equivalent to picking $u_4$ at the location of maximum wake strength. The near-wake length $x_0$ is also taken to be the $x$ position where $u_4$ is sampled because this inflection point marks the onset of the turbulent wake where wake recovery begins. The wake pressure $p_4$ is also calculated at the near-wake length $x = x_0$. Following Shapiro et al.[26], the lateral velocity $v_4$ is taken as its maximum, which is closer

to the actuator disk, approximately at $x/D = 0.5$ (see discussion by Shapiro et al.[26]). The same procedure for extracting the wake properties $u_4$, $v_4$, $x_0$, and $p_4$ is also used for wind turbine wakes in ABL inflows.

### Reporting summary

Further information on research design is available in the Nature Portfolio Reporting Summary linked to this article.

## Data availability

All data and figure-generating code is available as an open-source Python library at https://doi.org/10.5281/zenodo.10524066[81]. Source data are provided with this paper.

## Code availability

A reference implementation of the Unified Momentum Model is available as an open-source Python library at https://doi.org/10.5281/zenodo.10524066[81]. A blade element momentum implementation using the Unified Momentum Model is available at 10.5281/zenodo.11175618[82]. The large eddy simulation software source code, PadéOps[71], is available at https://github.com/Howland-Lab/PadeOps. The Unified Momentum Model is also available open-source on Github (https://github.com/Howland-Lab/Unified-Momentum-Model). The blade element momentum model based on the Unified Momentum Model is open-source on Github (https://github.com/Howland-Lab/MITRotor).

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

## Acknowledgements
K.S.H. and M.F.H. acknowledge funding from the National Science Foundation (Fluid Dynamics program, grant number FD-2226053, Program Manager: Dr. Ronald D. Joslin). J.L. acknowledges support from Siemens Gamesa Renewable Energy. K.S.H. additionally acknowledges funding through a National Science Foundation Graduate Research Fellowship under grant no. DGE-2141064. Simulations were performed on the Stampede2 and Stampede3 supercomputers under the NSF ACCESS project ATM170028.

## Author contributions
M.F.H. conceived the research. M.F.H., J.L., and K.S.H. developed the Unified Momentum Model. J.L. developed the Unified Momentum Model code. J.L. and M.F.H. analysed the data. M.F.H. and K.S.H. performed large eddy simulations. All authors contributed to manuscript writing and edits.

## Competing interests
The authors declare no competing interests.
