## [Peer Review File · Nature Communications]

Unified momentum model for rotor aerodynamics across operating regimesREVIEWER COMMENTS

Reviewer #1 (Remarks to the Author):

General comments

This article aims at developing a more generalised momentum approach for modelling rotor aerodynamics that is applicable to rotor yaw and thrust coefficients that are outside of the regime that is applicable to the traditional momentum approaches developed in the past and still used today. The proposed methodology builds upon more recent work dealing with momentum model generalisations across different operating regime.

- Overall, the proposed approach provides a step forward in the state of the art as it unifies various models to account for the more intricate physics persistent with rotor aerodynamics that classical methods do not take into account
- The authors still need to clarify the nature of the noted instabilities found at high thrust coefficients which pertain to the pressure calculation in the wake.
- There is some degree of uncertainty associated with the chosen parameters such as the shear layer growth rate which seems to require fine tuning. The effects on generalisability to different yaw regimes is still not fully addressed. Refer to the specific comments for some recommendations.

Please refer to the specific comments below that need to be addressed.

Specific comments

- Pg 2, please make use of the original references other than reference (7).
- The authors mention the instability associated with the pressure field calculation at high thrust coefficients. This reviewer agrees with the authors in attributing this issue to the Poisson's equation as implemented here. Nevertheless, the heuristic provided by setting an upper bound on the pressure drop is not entirely satisfactory vis a vis the ambitious aim that this work addresses. The authors should clarify whether they feel that such an issue causes a loss of generality in the upper thrust regime and if so, determine a limit to the stability of the model to higher thrust coefficients.
- It is also not entirely clear to this reviewer whether these instabilities are of a purely mathematical nature or whether there are subtle physical aspects that cannot be captured by the adoption of the Poisson's equation. This should be clarified.
- Can the authors clarify the resulting uncertainty in β for the yawed case? It seems that the authors did not optimise this parameter for the yawed situation and hence it is difficult to see how this could affect the reliability of the model in such cases. From the comparison's made with the high fidelity model it does seem that the unoptimized β for the yawed scenarios does not heavily influence the results but for the sake of generality the authors should be more thorough here. It is proposed to generalise the optimisation of β or otherwise provide a more convincing discussion on the effects on the yawed results.
- Figure 2 in supplementary information: The caption mentions a 10% shaded region but this does not seem to be visible. Is this a graphics issue or an omission?
- What are the author's considerations on the effects of radial flow and wake expansion especially in relation to the lateral velocity equation? Can they expand more on this in the main article?

Reviewer #2 (Remarks to the Author):

Please see the attachment.

A review of “**Unified Momentum Model for Rotor Aerodynamics Across Operating Regimes**”

by Jaime Liew, Kirby S. Heck, Michael F. Howland

Article No.: NCOMMS-24-02012, submitted to the Nature Communications

Editor: Dr. Gustavo Tontini

This manuscript presents a unified momentum model to efficiently predict power production and wake dynamics of wind turbine under given thrust coefficient and yaw angle. Compared to their previous work (Heck et al. 2023), the main contribution of the present manuscript is that the authors extend their previous work to high-thrust coefficient situations. This is done by updating the wake pressure recovery assumption with the pressure model proposed by Madsen (2023). The resultant model agrees well with large eddy simulations, indicating the usefulness of this treatment. Without doubt, this manuscript represents a significant progress in wind energy research. However, I don't think its novelty is high enough to be published in the Nature Communications.

General comments

1. Pressure model. For a yawed turbine, the flow is essentially three-dimensional or even unsteady if the thrust coefficient is very large. Thus, I don't understand why the pressure field can be obtained by solving the steady, two-dimensional Euler equations. Note that the spanwise velocity is obtained by the lifting line model, where the flow has already been assumed to be three-dimensional. In addition, the Bernoulli equation is valid only for steady and inviscid flow, why it can be used in the turbine wake region?
2. The present model is proposed under the uniform inflow condition. However, most wind turbines work in the atmospheric boundary layer where the inflow is essentially turbulent. So, what's the performance of the present model in realistic situations?
3. Even under inflow conditions, the model has only been compared with the actuator disk model simulations. However, the actuator disk model may not be appropriate for the simulations of yawed turbines. So, what's the performance of the present model compared to the simulations with more advanced turbine simulations (e.g. actuator line model)?

Minor comments

1. Page 11. “the proposed model had increasing error with increasing thrust coefficients because it assumed that the wake pressure recovers to freestream pressure”. The wake pressure recovery assumption is not the only reason for the poor performance. For example, Lu et al. (2023) found that under-prediction of the spanwise velocity also leads to the poor performance of the model.
2. Figure 2. “The shaded region corresponds to $\pm 10\%$ uncertainty in parameter β .” The shaded region is absent in all figures of the manuscript and the Supplementary Information (SI).

3. Figure 3. From the definition of Eq. (13), $a_n = 1$ means $u_d = 0$, i.e. the flow velocity is zero at the turbine location. Is this case meaningful in real world?
4. Figure 4. It could be better to also plot the red curves of unified momentum theory and LES in the same subfigure.
5. Equation (2) (and Eq. (2, 19, 20) in SI). In this equation all $\cos^2 \gamma$ should be $\cos \gamma$. Please double check the code and, if necessary, the model prediction results.
6. The sentence before Eq. (11) in SI. “Combing”?
7. Equation (14) in SI. The left term misses the volume integration.
8. Page 6, Line 4 in SI. s should be s/D .
9. Equation (37) in SI is different from Eq. (25) of Madsen (2023). Please clarify this difference.

References:

- [1] Heck, K. S., H. M. Johlas and M. F. Howland (2023). “Modelling the induction, thrust and power of a yaw-misaligned actuator disk.” *Journal of Fluid Mechanics* 959: A9.
- [2] Lu, J., C. Li, X. Li, H. Liu, G. Zhang, N. Liu and L. Liu (2023). “Analytical model for the power production of a yaw-misaligned wind turbine.” *Physics of Fluids* 35: 127109.
- [3] H. A. Madsen (2023). “An analytical linear two-dimensional actuator disc model and comparisons with computational fluid dynamics (CFD) simulations.” *Wind Energy Science* 8: 1853-1872.

Response to reviewer 1

Article title: Unified Momentum Model for Rotor Aerodynamics Across Operating Regimes

We thank the reviewer for their valuable and insightful feedback on our article. Point-to-point responses are listed below, and we have made the relevant changes to the revised manuscript. We believe that the revised manuscript is stronger and clearer than the initial submission.

Please find below our detailed responses to your comments (highlighted in blue). Additionally, please find a marked-up version showing all changes in the paper attached as a supplementary document.

General Comments

This article aims at developing a more generalised momentum approach for modelling rotor aerodynamics that is applicable to rotor yaw and thrust coefficients that are outside of the regime that is applicable to the traditional momentum approaches developed in the past and still used today. The proposed methodology builds upon more recent work dealing with momentum model generalisations across different operating regime.

Overall, the proposed approach provides a step forward in the state of the art as it unifies various models to account for the more intricate physics persistent with rotor aerodynamics that classical methods do not take into account. The authors still need to clarify the nature of the noted instabilities found at high thrust coefficients which pertain to the pressure calculation in the wake.

There is some degree of uncertainty associated with the chosen parameters such as the shear layer growth rate which seems to require fine tuning. The effects on generalisability to different yaw regimes is still not fully addressed. Refer to the specific comments for some recommendations.

We thank the reviewer for their time and detailed comments. We believe the revised manuscript is stronger and clearer. For readability, the specific aspects brought up in the General Comments are addressed below in the Specific Comments.

Specific Comments

Pg 2, please make use of the original references other than reference (7).

Thank you, we have revised the reference to refer to Rankine and Froude, who laid

the foundations of one-dimensional momentum theory.

The authors mention the instability associated with the pressure field calculation at high thrust coefficients. This reviewer agrees with the authors in attributing this issue to the Poisson's equation as implemented here. Nevertheless, the heuristic provided by setting an upper bound on the pressure drop is not entirely satisfactory vis a vis the ambitious aim that this work addresses. The authors should clarify whether they feel that such an issue causes a loss of generality in the upper thrust regime and if so, determine a limit to the stability of the model to higher thrust coefficients.

We appreciate the reviewer's important comment. The upper bound methodology is motivated by two findings in this study: 1) no analytical solution or computational relaxation was found that eliminates the instability for the required nonlinear pressure term; 2) large eddy simulations yield instability in this regime as shown in the Supplementary Information, which suggests that in fact, the flow itself is unstable in high thrust regimes (further discussed in the next comment). We agree that the heuristic upper bound on the pressure drop is not as satisfying as an analytical solution, but short of solving a full computational fluid dynamics calculation, no other approach investigated by the authors was as accurate nor as reliable.

The objective of this study is to develop an analytical set of equations that describe actuator disk aerodynamics across yaw misalignment and thrust coefficient regimes, for which there is no existing first-principles theory. This aim is motivated by the need to understand (i.e. interpret) and predict rotor performance across these regimes faster than computational fluid dynamics, as indicated by the widespread use of classical momentum theory. The derived analytical equations are first validated using *a priori* analysis, where the model is given the benefit of knowledge of the wake pressure. Then, the second task is to develop a prognostic equation for the pressure. In our investigation, we found that including the nonlinear pressure contribution from advection was imperative to accurately model the turbulent wake state (induction $a_n > 0.5$). Without the nonlinear pressure contribution, the model becomes entirely analytical and has no instability, but predictions exhibited significant deviations from large eddy simulations in this high thrust regime. Therefore, despite the relative complexity of the solution compared to other attributes of the model, we deemed it necessary to incorporate the nonlinear pressure term. This decision is supported by *a priori* and *a posteriori* analyses, which confirm the formulation of the Unified Model equations and validate the predictions of the pressure against large eddy simulations.

The method presented in this study provides a close approximation to the pressure while avoiding the adoption of a more complex method that would approach the computational expense of large eddy simulations. This discussion has been incorporated into the manuscript.

It is also not entirely clear to this reviewer whether these instabilities are of a purely mathematical nature or whether there are subtle physical aspects that cannot be captured by the adoption of the Poisson's equation. This should be clarified.

The instabilities stem from both numerical and physical factors and are particularly pronounced for induction factors $a_n > 0.7$. We diligently explored various strategies to mitigate these instabilities, including convergence enhancement techniques, dynamic mode decomposition analysis, adjustments to grid resolution and boundary conditions, as well as the introduction of numerical damping. More detail on our investigated solution methods to the instability challenge has been added to the manuscript. On the physical nature of the instabilities, in the Supplementary Information, we investigate the thrust coefficients (induction factors) that lead to unsteadiness in the large eddy simulations. The transition to unsteadiness occurs at approximately $C_T \approx 1.3$ ($a_n \approx 0.7$) which is where numerical instabilities are also encountered in the Poisson solution. To be clear, this is a correlative analysis, not causal, because the large eddy simulation and Euler equation solution differ significantly. It is however insightful to identify that the unstable regime is coincident with the unsteady regime in large eddy simulation.

After this thorough investigation, we developed the solution approach described in the manuscript. This approach offers an approximation of the nonlinear pressure field, serving as a conservative estimate or a lower bound before the onset of numerical instability. Again, the nonlinear pressure contribution could be neglected to result in no instabilities and a fully analytical model which yields lower error than classical momentum theory for all investigated regimes (see Figure 6 in the Supplementary Information, reproduced below as Figure R1), but this linear model is less accurate for $a_n > 0.7$ than the model that includes nonlinear pressure contributions. This decision is justified in the *a priori* analysis of the Unified Model equations and the model validation, affirming the adequacy of the lower bound approximation in yielding accurate results from the model.

Another potential avenue could involve conducting Reynolds-averaged Navier-Stokes (RANS) simulations on the system. Turbulent diffusion will likely lower the incidence of the instabilities. However, the underlying design philosophy in our work is aimed at striking a balance between the simplicity of 'momentum theory-like' analytical equations and the complexity of computational fluid dynamics. Consequently, we made a deliberate choice to maintain simplicity and interpretability in our approach. This results in a final model that is evaluated in microseconds on a laptop computer.

Figure R1: (a) Coefficient of thrust C_T , (b) Coefficient of power C_P , (c) streamwise wake velocity u_4 , and (d) density-normalized wake pressure deficit $(p_1 - p_4)/\rho$ as a function of the local thrust coefficient C'_T . The variables estimated by the model equations proposed here (Eq. 1), are shown. The shaded region corresponds to $\pm 10\%$ uncertainty in β . Results from LES, and classical one-dimensional momentum modeling are shown as a reference. The Betz limit of $C'_T = 2$ and $C_P = 16/27$ is also shown by the dashed green line. The model predictions from Steiros & Hultmark (2018) are also shown, as well as the Unified Momentum Model without the nonlinear pressure term.

Can the authors clarify the resulting uncertainty in β for the yawed case? It seems that the authors did not optimise this parameter for the yawed situation and hence it is difficult to see how this could affect the reliability of the model in such cases. From the comparison's made with the high fidelity model it does seem that the unoptimized β for the yawed scenarios does not heavily influence the results but for the sake of generality the authors should be more thorough here. It is proposed to generalise the optimisation of β or otherwise provide a more convincing discussion on the effects on the yawed results.

The single free parameter β in the model influences the near-wake length x_0 and has a well-accepted range in the literature, as discussed in the initial submission. In this study, we calibrate the model to the yaw-aligned LES data, and therefore, modeling results for yaw misalignment are predictions that are out-of-sample from the calibration. This is a useful trait because it reveals the generalization capabilities of the unknown parameter. The model results for x_0 for yaw aligned (calibration) and yaw misaligned (out-of-sample predictions) are shown in Figure R2, where the low error is incurred in the yawed cases. We can also calibrate our model to include the yaw-misaligned data. When calibrating β to particular yaw-misalignment values, we find that $\beta \in [0.1269, 0.1645]$. Considering all values of yaw-misalignment together, an overall least-squares estimate yields $\beta = 0.1456$, which is very close to the value of $\beta = 0.1409$ used in the paper. One advantage of using β as the free parameter in the Unified Momentum Model is that previous literature has reported a relatively narrow range of values for this proportionality constant. We have added additional discussion and new results of model comparison to LES in yaw-misaligned conditions to the Supplementary Information.

Figure R2: Near-wake length measured in LES and predicted by the near-wake length model (Eq. 4) plotted against (a) induction factor and (b) thrust coefficient C'_T for different yaw misalignment angles.

Figure 2 in supplementary information: The caption mentions a 10% shaded region but this does not seem to be visible. Is this a graphics issue or an omission?

The 10% error bars are indeed present in Figure 2. The error bars are not prominently visible due to the narrow range of the uncertainty they cause in the predictions. This very limited (visually imperceptible) uncertainty has been further explained in the caption of Figure 2.

What are the author's considerations on the effects of radial flow and wake expansion especially in relation to the lateral velocity equation? Can they expand more on this in the main article?

Thanks for noting this. These points were discussed in the Supplementary Information and we have added a discussion of them to the main article. To briefly summarize, the radial flow at the actuator disk is present, as shown in Figure 10 of the Supplementary Information. Because the velocity, including the radial flow, is continuous over the porous actuator disk, the radial flow cancels from the Bernoulli equation. Regarding the wake expansion in relation to the lateral velocity equation, the lateral velocity is accounted for in the mass and momentum conservation arguments used to derive the Unified Momentum Model, and therefore the wake expansion is consistent with the lateral velocity. As described in the Supplementary Information, the wake expansion is modeled using the area at station 4 (downwind wake location),

$$A_4 = \frac{(1 - a_n) \cos(\gamma) A_d u_\infty}{u_4}, \quad (1)$$

where u_4 and a_n are solved for using the Unified Momentum Model (Eqs.(1)-(5)), γ is the yaw, A_d is the actuator disk area, and u_∞ is the freestream velocity.

Response to reviewer 2

Article title: Unified Momentum Model for Rotor Aerodynamics Across Operating Regimes

We thank the reviewer for their valuable and insightful feedback on our article. Point-to-point responses are listed below, and we have made the relevant changes to the revised manuscript. We believe that the substantially revised manuscript is stronger and clearer than the initial submission.

To address the reviewer comments, we have performed two major new developments and added substantial new results. First, we extend our results to compare to more advanced turbine simulations (e.g. actuator line model). To do so, we couple the Unified Momentum Model with a blade element model, resulting in a new, first-principles blade element momentum (BEM) model which does not rely on empirical corrections. The BEM approach, widely used in wind and hydrokinetic energy, models rotors by coupling classical one-dimensional momentum theory to compute induction (momentum part) and blade element modeling to compute forces on rotating airfoils (blade element part). In our novel BEM method, the Unified Momentum Model directly replaces the current state-of-the-art in momentum modeling, which uses classical one-dimensional momentum theory with a high thrust correction and a skewed wake correction. The first-principles BEM model is compared to blade-resolved wind turbine simulations (ExaWind) generated by the National Renewable Energy Laboratory and baseline empirical BEM methods, demonstrating a reduction in predictive error ($\times 3$ error reduction for yaw of 30°) without relying on empiricism.

Second, we perform a comprehensive suite of large eddy simulations of an actuator disk immersed in turbulent atmospheric boundary layer flow, performing 70 independent large eddy simulations. The Unified Momentum Model is compared to this wide suite of large eddy simulations of the atmospheric boundary layer, demonstrating increased differences relative to uniform flow when both the thrust coefficient and yaw angle are higher, but the error is lower than the existing momentum theory widely used in the wind energy community. Existing BEM and analytical wake models are based on classical momentum theory that assumes the flow is uniform in the derivation, but these methods are widely applied in turbulent flows (i.e. analytical wake models use wake initial conditions of $\delta u_0 = 2a$). This is also the initial approach used in the Unified Momentum Model development since adding turbulence into the derivation challenges the development of analytical theory. We anticipate that the new Unified Momentum Model that does not use empirical corrections provides a new, solid basis to incorporate the atmospheric effects of turbulence and wind shear in future work.

In sum, we have made substantial revisions to the manuscript. Please find below our detailed responses to your comments (highlighted in blue). Additionally, please find a marked-up version showing all changes in the paper attached as a supplementary document.

This manuscript presents a unified momentum model to efficiently predict power production and wake dynamics of wind turbine under given thrust coefficient and yaw angle. Compared to their previous work (Heck et al. 2023), the main contribution of the present manuscript is that the authors extend their previous work to high-thrust coefficient situations. This is done by updating the wake pressure recovery assumption with the pressure model proposed by Madsen (2023). The resultant model agrees well with large eddy simulations, indicating the usefulness of this treatment. Without doubt, this manuscript represents a significant progress in wind energy research. However, I don't think its novelty is high enough to be published in the Nature Communications.

Thank you for your time and feedback. As the reviewer noted, this study develops and validates a model that accounts for both yaw misalignment and high thrust coefficient situations, both of which are currently modeled with empiricism in state-of-the-art wake modeling and BEM modeling applications. Accurate and reliable theory for the high thrust coefficient situation has eluded the aerodynamics community for a century, since Glauert's first empirical approach in 1926 [1]. Based on the reviewer's suggestions, we have also substantially expanded the scope of this paper by developing and evaluating a novel blade element momentum (BEM) model using the Unified Momentum Model for the induction closure (momentum component of BEM). The first-principles BEM developed here is compared to advanced turbine simulations, and represents the first BEM model that can predict yaw and high thrust coefficient operation without empirical corrections.

Both wake modeling and BEM are widely used by academic researchers and industry, e.g. NREL's BEM code OpenFAST [2] has thousands of references and similar growth is occurring for the newer FLORIS wake modeling code [3]. But the models underlying these codes currently rely on empiricism. In wind power textbooks, classical momentum theory is typically the first aspect introduced for wind turbine aerodynamics [4, 5]. Given the lack of theory beyond the classical model, this is usually subsequently followed by a discussion of the yaw and thrust coefficient regimes where classical theory breaks down, and empirical approaches ('corrections') are then introduced to handle this breakdown [4, 5]. The first-principles modeling in this study improves both our understanding and modeling capabilities of wind power aerodynamics, to the extent that the model is already being deployed in both OpenFAST and FLORIS at the request of the code developers. Based on these arguments, we believe this study is of high enough novelty to be published in Nature Communications.

General comments

Pressure model. For a yawed turbine, the flow is essentially three-dimensional or even unsteady if the thrust coefficient is very large. Thus, I don't understand why the pressure field can be obtained by solving the steady, two-dimensional Euler equations. Note that the spanwise velocity is obtained by the lifting line model, where the flow has already been

assumed to be three-dimensional. In addition, the Bernoulli equation is valid only for steady and inviscid flow, why it can be used in the turbine wake region?

The goal of this study is to develop a model for rotor aerodynamics with a similar computational cost to classical momentum theory. The purpose of this goal is two-fold: 1) a simple model for complex rotor aerodynamics improves interpretability and understanding of the key physical mechanisms that dictate rotor performance and therefore wind power extraction; 2) very fast computational modeling of induction, thrust, power, and wakes are necessary for wake modeling and BEM applications. The *a priori* analysis presented in the paper validates the form of the model when pressure is known, but an additional prognostic equation is needed to predict the wake pressure. In general, the flow and pressure fields may be three-dimensional and potentially unsteady in the wake, but the solution to a general three-dimensional, unsteady pressure field requires computational fluid dynamics. So instead, we seek to develop a prognostic equation that can be solved very efficiently to approximate the true pressure.

The proposed model importantly incorporates a prediction of the near-wake length (x_0 , Eq. 4, illustrated in Figure R1), which can be understood as the flow region that is dominated by inviscid effects [6, 7]. For $x > x_0$, the flow is turbulent and unsteady and the Bernoulli equation is not valid, but for $x < x_0$, the inviscid effects dominate and the Bernoulli equation is an appropriate modeling approximation of the true dynamics. The fidelity of the Bernoulli equation was discussed in the initial manuscript and is validated using *a priori* analysis in the Supplementary Information (Supplementary Figure 11, Supplementary section: **Streamtube analysis and budgets**). As the reviewer stated, as the thrust coefficient increases, the flow becomes more unsteady and the Bernoulli equation is not appropriate in the far wake ($x > x_0$). This is already addressed in the model, as the near-wake length x_0 is reduced as the thrust coefficient is increased, such that $x_0 = 0$ for $C_T = 2$ and $a_n = 1$, as shown in the figure below.

In summary, both the use of a two-dimensional Euler equation as a prognostic equation for the pressure and the use of the Bernoulli equation (only in the near-wake for $x < x_0$) are engineering modeling approximations. These modeling choices are extensively validated in the manuscript through *a priori* and *a posteriori* analysis and comparisons to data from large eddy simulations, demonstrating that the model yields very low error across all relevant yaw misalignment and thrust coefficient regimes relevant to wind power. Specifically, the model yields lower error than any existing model for all conditions considered. Future work may consider very efficient solutions to the three-dimensional, unsteady Navier-Stokes equations to model a three-dimensional pressure field, for example through surrogate modeling or physics-informed machine learning.

Figure R1: Near-wake length measured in LES and predicted by the near-wake length model (Eq. 4) plotted against (a) induction factor and (b) thrust coefficient C'_T for different yaw misalignment angles.

The present model is proposed under the uniform inflow condition. However, most wind turbines work in the atmospheric boundary layer where the inflow is essentially turbulent. So, what's the performance of the present model in realistic situations?

To address this question, we have added a substantial new suite of simulations and model results for a wind turbine operating in the turbulent, conventionally neutral atmospheric boundary layer (ABL). These new results represent 70 independent large eddy simulations (LES) of the turbulent ABL. The full details of the ABL LES simulations are provided in the Supplementary Information and are discussed more concisely in the Methods section of the main text. We have reproduced the key figure added to the main text below (Figure R2). Overall, the Unified Model predictions of the rotor characteristics (induction and thrust) and the wake velocities are in good agreement with ABL LES data. In general, model errors are greater in ABL inflow than for uniform inflow, particularly at the coincidence of high yaw-misalignment angles and thrust coefficients. Still, the Unified Momentum Model lowers prediction error across these yaw and thrust coefficient regimes by 60%, 83%, and 78% for the induction, streamwise wake velocity, and spanwise wake velocity, respectively, compared to classical one-dimensional momentum theory. This substantial new set of methods, results, and discussion have been added to the main text and the Supplementary Information.

While turbulence and other factors introduce significant complexity to realistic wind turbine systems, engineering models, such as low fidelity wake modeling and mid-fidelity aeroelastic turbine simulators based on BEM, are based on classical momentum theory in uniform inflow [8–11]. The uniform inflow case with high thrust coefficient has remained without a simple quantitative explanation based on first principles for almost 100 years since Glauert first introduced the empirical high thrust correction. The authors maintain that a focus on the canonical uniform inflow case provides a deeper understanding of the physics present in both yawed and high-thrust rotors. We also expect that the new Unified Momentum Model that reliably predicts yawed and high thrust coefficient operation without empiricism is the appropriate basis to model the more complex effects of wind shear and turbulence. Without this study, the starting point for theory developments would be to use classical momentum theory with the addition of empirical corrections for yaw and high thrust, but the degree to which those empirical corrections have been unintentionally biased by turbulence or wind shear effects in the underlying data is unknown to the authors.

Figure R2: Model comparison for rotor and near-wake properties in uniform and atmospheric boundary layer inflow as a function of local thrust coefficient C_T' and yaw γ . Subfigures show (a, b) rotor-normal induction factor, (c, d) thrust coefficient C_T , (e, f) initial streamwise velocity deficit $\delta u_0 = u_\infty - u_4$, and (g, h) initial lateral velocity deficit $\delta v_0 = v_\infty - v_4$. Model predictions are compared with large eddy simulations in (a, c, e, g) uniform inflow and (b, d, f, h) conventionally neutral atmospheric boundary layer conditions.

Even under inflow conditions, the model has only been compared with the actuator disk model simulations. However, the actuator disk model may not be appropriate for the simulations of yawed turbines. So, what's the performance of the present model compared to the simulations with more advanced turbine simulations (e.g. actuator line model)?

The authors agree with the reviewer about the importance of contextualizing the Unified Momentum Model's performance in relation to advanced rotor models. To compare momentum modeling to advanced turbine simulations (e.g. actuator line), we require modeling the aerodynamics at the blade level, for which the standard is blade element momentum (BEM) modeling (see [11]). To address this question, the authors have performed substantial additional research. We leverage the Unified Momentum Model as the basis for a novel BEM rotor model. This newly developed BEM model is compared against a blade-resolved simulation of a wind turbine performed by the National Renewable Energy Laboratory (NREL) using the ExaWind stack [12]. The novel BEM model based on the Unified Momentum Model lowers error by $\times 3$ for a yaw of 30° compared to classical momentum theory. This substantial new set of modeling methods, results, and discussion have been added to the main text and the Supplementary Information.

We have reproduced the key figure added to the main text below. Using a BEM model dependent on classical one-dimensional momentum theory without a high-thrust correction (Figure R3(a)) fails to converge in regions where the thrust coefficient exceeds unity, rendering its solution undefined. Consequently, the global optimal operating point using this classical model lies on the convergence boundary, preventing the accurate determination of the optimal controller set point. By incorporating an empirical high-thrust correction (Figure R3(b)), the upper-left quadrant of the coefficient of power surface can be realized, allowing a global optimal set point to be retrieved, but it requires empiricism to achieve, yielding uncertainty. Replacing the classical momentum theory and empirical high-thrust correction with the Unified Momentum Model (Figure R3(c)) enables the prediction of the coefficient of power across all of the relevant pitch and tip-speed ratio regimes, and therefore the identification of the optimal control strategy, without any empirical corrections. The results demonstrate that the Unified Momentum Model provides a new, robust foundation for BEM without requiring the empirical corrections (skewed wake correction, high thrust correction) upon which all current BEM models rely [2, 10, 11].

Figure R3: Contour plots showing the variation of power coefficient (C_P) with blade pitch angle (θ_p) and blade tip speed ratio (λ) using different thrust-induction momentum modeling closures in a blade element momentum (BEM) model implementation. Fully aligned (a, b, c) and yaw misaligned (d, e, f) conditions at $\gamma = 30^\circ$ are considered. The momentum modeling closure used in the BEM includes classical momentum theory (a), classical momentum theory with an empirical high-thrust correction (b), and the Unified Momentum Model (c), all incorporating Prandtl tip and root correction [13] in the blade element model. No empirical corrections are used in the Unified Momentum Model for induction.

Minor comments

Page 11. “the proposed model had increasing error with increasing thrust coefficients because it assumed that the wake pressure recovers to freestream pressure”. The wake pressure recovery assumption is not the only reason for the poor performance. For example, Lu et al. (2023) found that under-prediction of the spanwise velocity also leads to the poor performance of the model.

Thank you for this comment, and for mentioning the interesting results of Lu et al. (2023), which is cited in our main text. Lu et al. (2023) mentioned an underprediction of lateral velocity by the model of Heck et al. (2023) in their text and proposed an empirical correction by multiplying the v_4 equation with an empirical factor of 1.5. It does not appear to the authors that Lu et al. (2023) demonstrated an underprediction of the lateral velocity directly (i.e. no lateral velocity results are shown in the paper of Lu et al. (2023)). Here, we quantitatively evaluate the empirical correction of Lu et al. (2023). The empirical correction does not improve the overall model performance when considering all model output variables, as shown in Figure R4. Specifically, the empirical correction increases error in the lateral velocity v_4 , shown in Figure R4(a).

Lu et al. (2023) do not mention what metric they use to empirically adjust the equation for lateral velocity $v_4 = M \times -\frac{1}{4}C_T'(1 - a_n)^2 \sin(\gamma) \cos^2(\gamma)u_\infty$, where M is the empirical parameter proposed by Lu et al. (2023). If we calibrate the Unified Model by reducing the least-squares error between the modeled v_4 and LES v_4 extracted from uniform inflow LES data, we find that an empirical correction very near unity ($M = 0.96$), suggesting that v_4 should not be corrected. Interestingly, we find that if we choose to calibrate this empirical factor by minimizing the least-squares error in C_P between LES and the model predictions, then the empirical correction factor exceeds unity ($M = 1.35$). But as is shown by Figure R4, the very marginal improvement in C_P at high yaw misalignment angles from the empirical calibration ($M = 1.35$) increases error in the model prediction of v_4 when compared with LES. In staying consistent with the physics-based derivation and reducing the number of empirical parameters in the Unified Model, we believe that the equation for v_4 as it is currently presented in the main text best represents the physics of the yawed actuator disk. Further, for wake modeling applications, it is critical to accurately predict the wake velocities, beyond only predicting C_P .

Figure R4: Comparison between the Unified Momentum Model and the Unified Momentum Model empirically modified v_4 increased by a factor of 3/2 in the lateral velocity equation (Lu *et al.* modified v_4), compared with uniform inflow LES data, showing a) comparison of initial lateral velocity deficit $\delta v_0 = -v_4$ and b) coefficient of power C_P .

Figure 2. “The shaded region corresponds to $\pm 10\%$ uncertainty in parameter β .” The shaded region is absent in all figures of the manuscript and the Supplementary Information (SI).

For Figure 2, and all other figures in the initial submission, the uncertainty range was indeed present in the plotting. The lack of a visual signature in some plots actually stems from the (visually) negligible impact of uncertainty in the parameter β , which was the key point of this uncertainty investigation. We agree that this was inadequately clear in the initial submission. Regarding the physical mechanisms for the degree of uncertainty observed, Figure 2 illustrates the axial induction factor a_n and the thrust coefficient C'_T . The uncertainty range was indeed included in the plotting process, however, the range is negligible for these particular variables. This is because a_n and C'_T are relatively insensitive to β , as properties at the rotor. We find that quantities related to the wake, such as the wake velocity and far-wake pressure p_4 (Figure 3), and the near wake length x_0 (shown in the Supplementary Information), are more sensitive to β . We have further clarified the shaded region and the parameter uncertainty results in the figure captions.

Figure 3. From the definition of Eq. (13), $a_n = 1$ means $u_d = 0$, i.e. the flow velocity is zero at the turbine location. Is this case meaningful in real world?

While wind turbines rarely operate at the condition where the axial induction factor a_n equals 1, this scenario is relevant for general propeller analysis. At and above induction of unity ($a_n \geq 1$), the wake flow is separated immediately downwind (i.e. $x_0 = 0$), resulting in the bluff body wake of a flat circular plate [14]. In such cases, the presented model is no longer valid, because flow separation immediately occurs ($x_0 = 0$ in the model). While the regime of unity induction ($a_n \geq 1$) is likely not often relevant to wind power applications, it is useful to connect this Unified Momentum Model with research focused on bluff body wake dynamics [15, 16]. Bluff body dynamics are an unsolved problem and require computational fluid dynamics to resolve.

Figure 4. It could be better to also plot the red curves of unified momentum theory and LES in the same subfigure.

Thank you for the suggestion. Figure 4 has been updated to incorporate the power-maximizing operating points obtained from LES, plotted alongside the setpoints predicted by the Unified Momentum Model and one-dimensional momentum theory. This gives a more direct illustration of the differences between these approaches.

Equation (2) (and Eq. (2, 19, 20) in SI). In this equation all $\cos^2 \gamma$ should be $\cos \gamma$. Please double check the code and, if necessary, the model prediction results.

Thank you for checking the equations in detail, but the equations in the original

manuscript main text and SI were correct. As the reviewer notes, Equation (2) in the main text is the same as Equation (20) in the SI where it is derived. The derivation of Eq. 19 comes from Eq. 18 which is from integral streamwise momentum conservation combined with conservation of mass. Restating Eq. 18 here:

$$\vec{F}_T \cdot \hat{i} = \rho(1 - a_n) \frac{u_\infty}{u_4} \cos(\gamma) A_d (u_4^2 - u_\infty u_4) + \frac{(1 - a_n) \cos(\gamma) A_d u_\infty}{u_4} (p_4 - p_1). \quad (1)$$

The left hand side contains the thrust force, which is given by

$$\vec{F}_T = -\frac{1}{2} \rho C'_T A_d (1 - a_n)^2 \cos^2(\gamma) u_\infty^2 [\cos(\gamma) \hat{i} + \sin(\gamma) \hat{j}]. \quad (2)$$

Plugging in the thrust force and taking the dot product with the \hat{i} direction yields

$$-\frac{1}{2} \rho C'_T A_d (1 - a_n)^2 \cos^3(\gamma) u_\infty^2 = \rho(1 - a_n) \frac{u_\infty}{u_4} \cos(\gamma) A_d (u_4^2 - u_\infty u_4) + \frac{(1 - a_n) \cos(\gamma) A_d u_\infty}{u_4} (p_4 - p_1). \quad (3)$$

Note that an extra cosine appears on the left hand side because of the dot product between the thrust force vector \vec{F}_T with the \hat{i} direction. Simplifying this equation yields

$$u_4^2 + \left(\frac{1}{2} C'_T (1 - a_n) \cos^2(\gamma) u_\infty - u_\infty \right) u_4 + \frac{1}{\rho} (p_4 - p_1) = 0, \quad (4)$$

which is the Eq. 19 in the original SI. Using the quadratic equation yields

$$u_4 = -\frac{1}{4} C'_T (1 - a_n) \cos^2(\gamma) u_\infty + \frac{u_\infty}{2} + \frac{1}{2} \sqrt{\left(\frac{1}{2} C'_T (1 - a_n) \cos^2(\gamma) u_\infty - u_\infty \right)^2 - \frac{4(p_4 - p_1)}{\rho}}, \quad (5)$$

which is both Eq. 2 and Eq. 20 in the original SI.

The sentence before Eq. (11) in SI. “Combing”?

We have fixed the typographical error and improved the notation in the equation.

Equation (14) in SI. The left term misses the volume integration.

Thank you for checking the equations in detail and for this catch. We have fixed the typographical error.

Page 6, Line 4 in SI. s should be s/D .

Thank you for pointing out the typographic errors. The necessary corrections have been made in the text.

Equation (37) in SI is different from Eq. (25) of Madsen (2023). Please clarify this difference.

Madsen (2023) presents their formulation normalized by rotor radius. We have chosen to remain consistent with the rest of the document by keeping the formulation normalized by rotor diameter, hence the difference in the coefficients of x and y . This nuance has been further explained in the text.

References

- [1] H. Glauert, “The analysis of experimental results in the windmill brake and vortex ring states of an airscrew,” Tech. Rep. 1026, ARCR R&M, 1926.
- [2] J. M. Jonkman, M. L. Buhl Jr, *et al.*, “Fast user’s guide,” *Golden, CO: National Renewable Energy Laboratory*, vol. 365, p. 366, 2005.
- [3] C. Bay, J. R. King, P. A. Fleming, L. Martinez, R. M. Mudafort, E. J. Simley, and M. J. Lawson, “Floris: A brief tutorial,” tech. rep., National Renewable Energy Lab (NREL), Golden, CO (United States), 2020.
- [4] T. Burton, N. Jenkins, D. Sharpe, and E. Bossanyi, *Wind energy handbook*. John Wiley & Sons, 2011.
- [5] D. M. Eggleston and F. Stoddard, *Wind turbine engineering design*. Van Nostrand Reinhold Co. Inc., New York, NY, 1987.
- [6] C. R. Shapiro, D. F. Gayme, and C. Meneveau, “Modelling yawed wind turbine wakes: a lifting line approach,” *J. Fluid Mech.*, vol. 841, 2018.
- [7] M. Bastankhah and F. Porté-Agel, “Experimental and theoretical study of wind turbine wakes in yawed conditions,” *J. Fluid Mech.*, vol. 806, pp. 506–541, 2016.
- [8] N. O. Jensen, “A note on wind generator interaction,” *Rep. Risø-M-2411*, vol. 2411, 1983.
- [9] M. Bastankhah and F. Porté-Agel, “A new analytical model for wind-turbine wakes,” *Renewable Energy*, vol. 70, pp. 116–123, 2014.
- [10] H. A. Madsen, T. J. Larsen, G. R. Pirrung, A. Li, and F. Zahle, “Implementation of the blade element momentum model on a polar grid and its aeroelastic load impact,” *Wind Energy Science*, vol. 5, no. 1, pp. 1–27, 2020.
- [11] P. J. Moriarty and A. C. Hansen, “Aerodyn theory manual,” tech. rep., National Renewable Energy Lab., Golden, CO (US), 2005.

- [12] M. A. Sprague, S. Ananthan, G. Vijayakumar, and M. Robinson, “Exawind: A multi-fidelity modeling and simulation environment for wind energy,” in *Journal of Physics: Conference Series*, vol. 1452, p. 012071, IOP Publishing, 2020.
- [13] L. Prandtl, “Applications of modern hydrodynamics to aeronautics,” 1923.
- [14] K. Steiros and M. Hultmark, “Drag on flat plates of arbitrary porosity,” *Journal of Fluid Mechanics*, vol. 853, p. R3, 2018.
- [15] H. Choi, W.-P. Jeon, and J. Kim, “Control of flow over a bluff body,” *Annu. Rev. Fluid Mech.*, vol. 40, pp. 113–139, 2008.
- [16] J. L. Ortiz-Tarin, S. Nidhan, and S. Sarkar, “The high-reynolds-number stratified wake of a slender body and its comparison with a bluff-body wake,” *Journal of Fluid Mechanics*, vol. 957, p. A7, 2023.

REVIEWER COMMENTS

Reviewer #1 (Remarks to the Author):

Dear authors,

I am satisfied with the careful replies provided. I also appreciate how the authors prioritised simplicity and efficiency of the model while at the same time answering the review comments diligently.

Reviewer #2 (Remarks to the Author):

Please see the attachment.

Reviewer #2 (Attachment):

A review of “**Unified Momentum Model for Rotor Aerodynamics Across Operating Regimes**”

by Jaime Liew, Kirby S. Heck, Michael F. Howland

Article No.: NCOMMS-24-02012A, submitted to the Nature Communications

Editor: Dr. Gustavo Tontini

Recommendation: Minor revision

The authors have performed two major new developments, i.e. a unified blade element momentum model and a comprehensive suite of large eddy simulations of an actuator disk immersed in turbulent atmospheric boundary layer flow. The good agreement of the model and simulations supports the validity of the model under arbitrary inflow angles and thrust coefficients. The novelty of the manuscript has thus been improved further than the previous version. However, the Prandtl tip and root correction was used in the unified blade element momentum model (Figure 5 in the Main Text). Thus, the model is not “without using corrections” (Abstract) as asserted by the authors. On the other hand, for the Unified Momentum Model derivation some physical assumptions are unclear and should be clarified more clearly (Supplementary Information). In addition, there are many other minor issues in both the Main Text and Supplementary Information that need to be explained, clarified or corrected. Based on these facts, I think the manuscript still needs to be revised further before it can be published in Nature Communications.

Below are the detailed comments that are in the order of page number rather than importance.

Comments on highlighted Main Text

1. Abstract. “The Unified Momentum Model is additionally coupled with a blade element model to enable blade element momentum modeling predictions of wind turbines in high thrust coefficient and yaw misaligned states without using corrections.” This is not true as the Prandtl tip and root correction was used (see Figure 5).
2. Page 6. “The Unified Momentum Model is coupled with blade element modeling to result in a new blade element momentum (BEM) modeling approach that can predict the effects of rotor misalignment and high thrust operation without empirical corrections.” This is not true as the Prandtl tip and root correction was used (see Figure 5).
3. Page 7. “Reynolds Averaged Navier Stokes” should be “Reynolds-Averaged Navier-Stokes”.
4. Page 9. Remove “CFD”.
5. Page 14. “We develop a first-principles blade element momentum model based on the Unified Momentum Model that predicts high thrust and yawed operation without empirical corrections

for the first time.” This is not true as the Prandtl tip and root correction was used (see Figure 5).

6. Page 15. “without or without corrections” should be “without or with corrections”.
7. Page 16. “the Unified Momentum Model yields a new approach to blade element momentum modeling that predicts yawed and high thrust operation from first-principles, without empirical corrections.” This is not true as the Prandtl tip and root correction was used (see Figure 5).
8. Page 17. “large eddy simulations” should be “LES”; “atmospheric boundary layer” should be “ABL”.
9. Pages 17-18. “The first-principles BEM model developed here based on the Unified Momentum Model accurately predicts the power and forces of a turbine in high thrust and yaw misalignment without empirical corrections.” This is not true as the Prandtl tip and root correction was used (see Figure 5).
10. Figure 3, Line 4. Remove “model equations”.
11. Figures 3 and 6. How are the values of p_4 , u_4 and v_4 calculated in LES? These are critical quantities in the Unified Momentum Model and thus it could be very helpful to state clearly how these quantities can be obtained in simulations and measurements.
12. References. Please double check references 26, 35, 52, 63, 69, 74, etc.
13. Page 32. The definition for F_T is given by Eqs. (7) and (9). Please consider to rephrase the text.

Comments on highlighted Supplementary Information

1. Pages 2, 4, 11, etc. F_T is a vector and thus there should always be an arrow at its top.
2. Page 3. u_4 in Eq. (11) is a vector including two components u_4 and v_4 . Here two u_4 could cause some misleading. In addition, it could be helpful to state clearly that the assumption $\mathbf{u}_2 = \mathbf{u}_3$ has been used when deriving Eq. (11).
3. Page 4, Eqs. (15) and (16). The control volume/surface and the underlying assumptions need to be specified: is it the circular cylinder with cross-sectional area A_4 or with arbitrary cross-sectional area? If it is the former, it seems that the streamwise velocity on the cylinder side surface is assumed as u_1 ; If it is the latter, the streamwise velocity and pressure on the downstream cylinder cross-sectional area (except A_4) is u_1 and p_1 . Please clarify this issue.

4. Page 4. Please specify the expression of \dot{m}_{out} .
5. Page 12. The second w_x should be w_y .
6. Page 15. The standard range of beta values in the literature discussed previously is not (0.116-0.154), but (0.154/2-0.154), see Page 14.
7. Page 15. "... and therefore since the parameter has only been optimized for ..." This sentence may have some grammar issue, please double check it.
8. Page 22. It could be helpful to specify the expression of the tip-loss correction $F(\mu)$.
9. Page 23, the third paragraph. Please specify which figure.
10. Page 25, Eqs. (78) and (79). The control surface needs to be specified: is it the circular cylinder with a very large cross-sectional area or the streamtube sketched in Fig. 1 of the Main Text? If it is the former, it seems that there should be an additional term in Eq. (79), i.e. $p_1 \dot{m}_{out}$, and thus the last two terms in Eq. (79) should be $(p_1 - p_4)A_4 u_4$; If it is the latter, the second term in Eq. (79) should be $A_1 p_1 u_1$. From my opinion I think it should be the former.
11. Page 26. It could be helpful to mention that Eq. (17) has been used to obtain Eq. (84).
12. Equation (85). The first term on the right-hand side should divide the cross-sectional area.
13. Figure 1. The legend for the color bar should be the pressure difference.
14. Figure 5. "streamwise" should be "Streamwise".
15. Figure 9. The cross and square symbols are not visible in figure 9b.
16. Figure 11b. This figure shows that the model residual of a yawed turbine without the pressure contribution is smaller than that with the pressure contribution. Any explanation for this model behavior?
17. References. Please double check references 4, 9, 26, 35, 41, etc.

Response to reviewer 2

Article title: Unified Momentum Model for Rotor Aerodynamics Across Operating Regimes

We thank the reviewer for their valuable comments on our article. Point-to-point responses are listed below, and we have made the relevant changes to the revised manuscript.

Please find below our detailed responses to your comments (highlighted in blue). Additionally, please find a marked-up version showing all changes in the paper attached as supplementary documents for both the main text and the Supplementary Information.

General Comments

The authors have performed two major new developments, i.e. a unified blade element momentum model and a comprehensive suite of large eddy simulations of an actuator disk immersed in turbulent atmospheric boundary layer flow. The good agreement of the model and simulations supports the validity of the model under arbitrary inflow angles and thrust coefficients. The novelty of the manuscript has thus been improved further than the previous version. However, the Prandtl tip and root correction was used in the unified blade element momentum model (Figure 5 in the Main Text). Thus, the model is not “without using corrections” (Abstract) as asserted by the authors. On the other hand, for the Unified Momentum Model derivation some physical assumptions are unclear and should be clarified more clearly (Supplementary Information). In addition, there are many other minor issues in both the Main Text and Supplementary Information that need to be explained, clarified or corrected. Based on these facts, I think the manuscript still needs to be revised further before it can be published in Nature Communications.

We thank the reviewer for their time and detailed comments. The revised manuscript is stronger and clearer. For readability, the specific aspects brought up in the General Comments are addressed below.

Comments on highlighted Main Text

1. Abstract. “The Unified Momentum Model is additionally coupled with a blade element model to enable blade element momentum modeling predictions of wind turbines in high thrust coefficient and yaw misaligned states without using corrections.” This is not true as the Prandtl tip and root correction was used (see Figure 5).

We agree and we have adjusted the manuscript at the relevant locations. The Unified Momentum Model allows for the modeling of both high thrust and yaw misaligned actuator disks without empirical corrections, but, there are still other assumptions involved

in transitioning from an actuator disk model to a blade element momentum model, particularly regarding the losses at the blade tips. We have revised the text to clarify that the Unified Momentum Model eliminates the use of corrections on the momentum theory side and that tip loss corrections are still applied in the blade element model when coupling the Unified Momentum Model to blade element modeling.

2. Page 6. “The Unified Momentum Model is coupled with blade element modeling to result in a new blade element momentum (BEM) modeling approach that can predict the effects of rotor misalignment and high thrust operation without empirical corrections.” This is not true as the Prandtl tip and root correction was used (see Figure 5).

See response to comment 1.

3. Page 7. “Reynolds Averaged Navier Stokes” should be “Reynolds-Averaged Navier-Stokes”.

Corrected.

4. Page 9. Remove “CFD”.

Corrected.

5. Page 14. “We develop a first-principles blade element momentum model based on the Unified Momentum Model that predicts high thrust and yawed operation without empirical corrections for the first time.” This is not true as the Prandtl tip and root correction was used (see Figure 5).

See response to comment 1.

6. Page 15. “without or without corrections” should be “without or with corrections”.

Corrected.

7. Page 16. “the Unified Momentum Model yields a new approach to blade element momentum modeling that predicts yawed and high thrust operation from first-principles, without empirical corrections.” This is not true as the Prandtl tip and root correction was used (see Figure 5).

See response to comment 1.

8. Page 17. “large eddy simulations” should be “LES”; “atmospheric boundary

layer” should be “ABL”.

Corrected.

9. Pages 17-18. “The first-principles BEM model developed here based on the Unified Momentum Model accurately predicts the power and forces of a turbine in high thrust and yaw misalignment without empirical corrections.” This is not true as the Prandtl tip and root correction was used (see Figure 5).

See response to comment 1.

10. Figure 3, Line 4. Remove “model equations”.

Corrected.

11. Figures 3 and 6. How are the values of p_4 , u_4 and v_4 calculated in LES? These are critical quantities in the Unified Momentum Model and thus it could be very helpful to state clearly how these quantities can be obtained in simulations and measurements.

We have further elaborated our methodology for extracting wake properties u_4 , v_4 , x_0 , and p_4 from the LES data in the Methods section and also clarified this in the Results section of the main text.

12. References. Please double check references 26, 35, 52, 63, 69, 74, etc.

Thank you for catching this. We have thoroughly reviewed the references and have updated any missing fields.

13. Page 32. The definition for F_T is given by Eqs. (7) and (9). Please consider to rephrase the text.

Thank you for the comment. We have modified the sentence which first mentions F_T to connect it with the definition in the following section.

Comments on highlighted Supplementary Information

1. Pages 2, 4, 11, etc. F_T is a vector and thus there should always be an arrow at its top.

Now corrected throughout the Supplementary Information.

2. Page 3. u_4 in Eq. (11) is a vector including two components u_4 and v_4 . Here

two u_4 could cause some misleading. In addition, it could be helpful to state clearly that the assumption $\mathbf{u}_2 = \mathbf{u}_3$ has been used when deriving Eq. (11).

Thank you for catching this notation, we agree, and have adjusted the vector notation to improve the clarity. As in classical momentum theory, the velocity is assumed to be continuous over the porous actuator disk such that $\vec{u}_2 = \vec{u}_3$. We have clarified this point in the manuscript.

3. Page 4, Eqs. (15) and (16). The control volume/surface and the underlying assumptions need to be specified: is it the circular cylinder with cross-sectional area A_4 or with arbitrary cross-sectional area? If it is the former, it seems that the streamwise velocity on the cylinder side surface is assumed as u_1 ; If it is the latter, the streamwise velocity and pressure on the downstream cylinder cross-sectional area (except A_4) is u_1 and p_1 . Please clarify this issue.

The control volume is the circular cylinder with an arbitrary cross-sectional area, as shown by the box with dashed lines in Figure 1 of the main text. As the reviewer indicated, the streamwise velocity and pressure on the downstream cylinder cross-sectional area (except A_4) is u_1 and p_1 . We have clarified this in the manuscript.

4. Page 4. Please specify the expression of \dot{m}_{out} .

We have specified the expression in the manuscript, which comes from conservation of mass in the control volume.

5. Page 12. The second w_x should be w_y .

We have corrected the manuscript accordingly.

6. Page 15. The standard range of beta values in the literature discussed previously is not (0.116-0.154), but (0.154/2-0.154), see Page 14.

The description of the standard β range has been updated accordingly.

7. Page 15. "... and therefore since the parameter has only been optimized for ...". This sentence may have some grammar issue, please double check it.

Corrected.

8. Page 22. It could be helpful to specify the expression of the tip-loss correction $F(\mu)$.

The definition of the Prandtl tip and root loss correction has now been included in the manuscript.

9. Page 23, the third paragraph. Please specify which figure.

The reference to Figure 8 has been changed to Figure 8(a-f).

10. Page 25, Eqs. (78) and (79). The control surface needs to be specified: is it the circular cylinder with a very large cross-sectional area or the streamtube sketched in Fig. 1 of the Main Text? If it is the former, it seems that there should be an additional term in Eq. (79), i.e. $p_1 \dot{m}_{\text{out}}$, and thus the last two terms in Eq. (79) should be $(p_1 - p_4)A_4 v_4$. If it is the latter, the second term in Eq. (79) should be $A_1 p_1 u_1$. From my opinion I think it should be the former.

Thank you for this question and for checking the equations in detail. As the reviewer has suggested, the control volume and surface are the same as in the main model development: the circular cylinder with a very large cross-sectional area (the ‘former’ control surface discussed in the reviewer’s comment). This is shown in Figure 1 of the main text as the box with dashed lines. We have fixed the term that the reviewer identified, thank you for checking in detail. The final answer (Eq. (88)) and agreement with the derivation from the Bernoulli equation (Eq. (89)) are unchanged. We have further clarified this in the manuscript.

11. Page 26. It could be helpful to mention that Eq. (17) has been used to obtain Eq. (84).

We have added this note to the manuscript.

12. Equation (85). The first term on the right-hand side should divide the cross-sectional area.

Thank you for catching this, we have adjusted the manuscript accordingly.

13. Figure 1. The legend for the color bar should be the pressure difference.

Thank you, we have corrected the color bar label.

14. Figure 5. “streamwise” should be “Streamwise”.

Corrected.

15. Figure 9. The cross and square symbols are not visible in figure 9b.

Thank you for pointing this out. Figure 9b indeed does not contain the LES and Unified Momentum Model results relating to RANS as these correspond exactly to the blue points. We have updated the figure to remove the incorrect thrust cosine fits for these values in Figure 9b.

16. Figure 11b. This figure shows that the model residual of a yawed turbine without the pressure contribution is smaller than that with the pressure contribution. Any explanation for this model behavior?

Thank you for this question. We have expanded the discussion of this result in the Supplementary Information and we briefly summarize here. The full energy budget and residuals are shown in Figure 11, demonstrating that the pressure contribution must be included to yield low residual. The model approximations are also shown in Figure 11. The difference between the *budget* and the *model* are that the model replaces the spatial average of the budget terms with terms that are instead evaluated with a spatial average of the field variable (e.g. $\frac{1}{A} \int_A u_4 dA$, where A is the streamtube cross-section). This change in the order of operations is an evaluation of the uniform flow assumption used in the derivation of both classical momentum theory and the Unified Momentum model.

For the yaw misaligned case of $\gamma = 40^\circ$, the streamtube averaged *budget residual* is smallest when including the pressure contribution. The pressure term decreases in magnitude when the disk is misaligned because the pressure difference is dependent on the thrust force, and therefore, for fixed C'_T , neglecting the wake pressure has a lower impact on both the budget and the model as the yaw magnitude increases. The higher *model residual* for yaw misaligned porous disks is related to the increasing three-dimensionality in the wake of a yaw misaligned turbine. Spatial variations in u_4 and p_4 are not captured by the *model*. The *budget* terms resolve the three-dimensionality. Going to a three-dimensional representation, as in the full *budget* shown in Figure 11, improves accuracy relative to the model, but would increase the complexity of the model form and challenge arriving at an analytical form. The model that neglects the wake pressure has a benefit of fortuitous error cancellation, since Term I (wake velocity u_4 contribution) and Term III (wake pressure p_4 contribution) have opposite sign.

17. References. Please double check references 4, 9, 26, 35, 41, etc.

Thank you for catching this. We have thoroughly reviewed the references and have updated any missing fields.

REVIEWERS' COMMENTS

Reviewer #2 (Remarks to the Author):

All my previous comments have been answered very satisfactorily by the authors. Therefore, I am very happy to recommend the publication of the manuscript in Nature Communications.

Reviewer #2 (Remarks on code availability):

None.